

# Formal modeling of a causal consistent distributed system and verification of its history *via* model checking using colored Petri net

Khalid Amjed Mohammed Alsaegg[1], Saeid Pashazadeh[2] and Mina Zolfy Lighvan[1]

[1] Department of Computer Engineering, Faculty of Electrical and Computer Engineering, University of Tabriz, Tabriz, East Azerbijan, Iran
[2] Department of Information Technology, Faculty of Electrical and Computer Engineering, University of Tabriz, Tabriz, East Azerbaijan, Iran

## ABSTRACT

Various consistency models for replicated distributed systems (DSs) have been developed and are usually implemented in the middleware layer. Causal consistency (CC) is a widely used consistency model appropriate for distributed applications like discussion groups and forums. One of the known distributed algorithms for CC is based on logical time synchronization with Fidge vector clocks that use the concepts of the hold-back and delivery queues for each replica. The basics of the algorithm and its assumptions are presented in the article. Then, a novel formal hierarchical colored Petri net model of a DS with CC support and three constituting replicas is presented. The proposed model operates based on the presented distributed algorithm for CC support with potential randomness for delays in message delivery. The article tries to answer the question: is a given distributed history (DH) a valid image of a causal-consistent distributed system (CCDS)? The proposed model validates a DH *via* model checking. The question is answered by the execution of the proposed model and the generation of its state space graph (SSG). Required model checking functions are developed for automatically analyzing SSG for (1) extracting the existence of the answer and (2) extraction of the shortest proof scenarios that can generate the given input DH. The model was used to analyze four case study examples. The article presents three effective techniques for decreasing the state space explosion problem. Results show that the colored Petri net model of a CCDS can automatically validate a DH using model checking.

# INTRODUCTION

A switch-based multicomputer system is a distributed system (DS) composed of a few independent computers connected *via* computer networks set up with switches (*Tanenbaum & Steen, 2006*). In this article, DS refers to switch-based multicomputer systems. Replication is one of the basic approaches for improving fault-tolerance,

Corresponding author
Saeid Pashazadeh,
s_pashazadeh@yahoo.com

availability and reliability of the DS, but replication is costly in terms of (1) hardware usage and (2) communication and processing overheads for keeping the replicas consistent. A consistency model specifies a contract between the data store and processes. The data store guarantees that if processes follow the operation rules on its data, the data store's reading, writing, or updating results will be predictable and consistent (*Tanenbaum & Steen, 2006*). The consistency model is an abstraction model on the DS generally implemented by middleware. Using a consistency model eases the development of distributed applications for programmers. Based on the DS and its applications, many categories of consistency models, namely, client-centric, data-centric, and a combination of both, have been developed (*Aldin et al., 2019*). Figure 1 shows the taxonomy of consistency models.

Data-centric consistency models are generally used in distributed shared memory, shared data centers, and distributed databases. Strict, sequential, linearizable, causal, First-In-First-Out (FIFO), weak, release, lazy release, and entry consistency are the most famous data-centric consistency models. Strict consistency is the ideal model for programmers because DS behaves like a single computer system under its support. All programmers are familiar with programming a single system, which is an easy task. Under a strict consistency model, each update will be applied immediately on all replicas. A strict consistency model is not achievable with the current technology of computers because of delay and packet loss in computer networks. Therefore, other consistency models have been developed, such as sequential and causal consistency (CC). A distributed application development by programmers needs familiarity and experience to work with a consistency model, which is sometimes tedious. CC is a weaker form of sequential consistency appropriate for applications like distributed discussion groups and distributed resource management.

Most people start programming by developing programs for a single computer system. Then, in advanced stages, they may program multi-process or multi-thread applications or develop programs for DSs. Programming distributed systems has many complications compared to programming single computer systems. Graduate students learn about these complexities in courses such as distributed systems, advanced operating systems, or distributed databases. One of these complications is the issue of maintaining the consistency of replicas in DS. Usually, managing replicas and maintaining their consistency is the responsibility of the middleware layer in DS. Various middlewares have been developed for various types of DSs like wireless sensor networks, *ad-hoc* networks, and cloud computing, each of which supports specific consistency models.

Normally, all graduate students are familiar with strict consistency because it mimics the behavior of a single computer system in a real DS, but implementing this type of consistency model is practically impossible. Therefore, weaker consistency models, such as causal or sequence consistency, have been developed depending on the type of application. Each of these types of consistencies is suitable for a specific range of applications and is designed according to the needs of those applications. Their implementation is easier due to their weaker guarantees compared to absolute compatibility.

The remaining sections of the article have been organized as follows: "Related Works" reviews previous studies on the application of Petri net and specially colored Petri net in

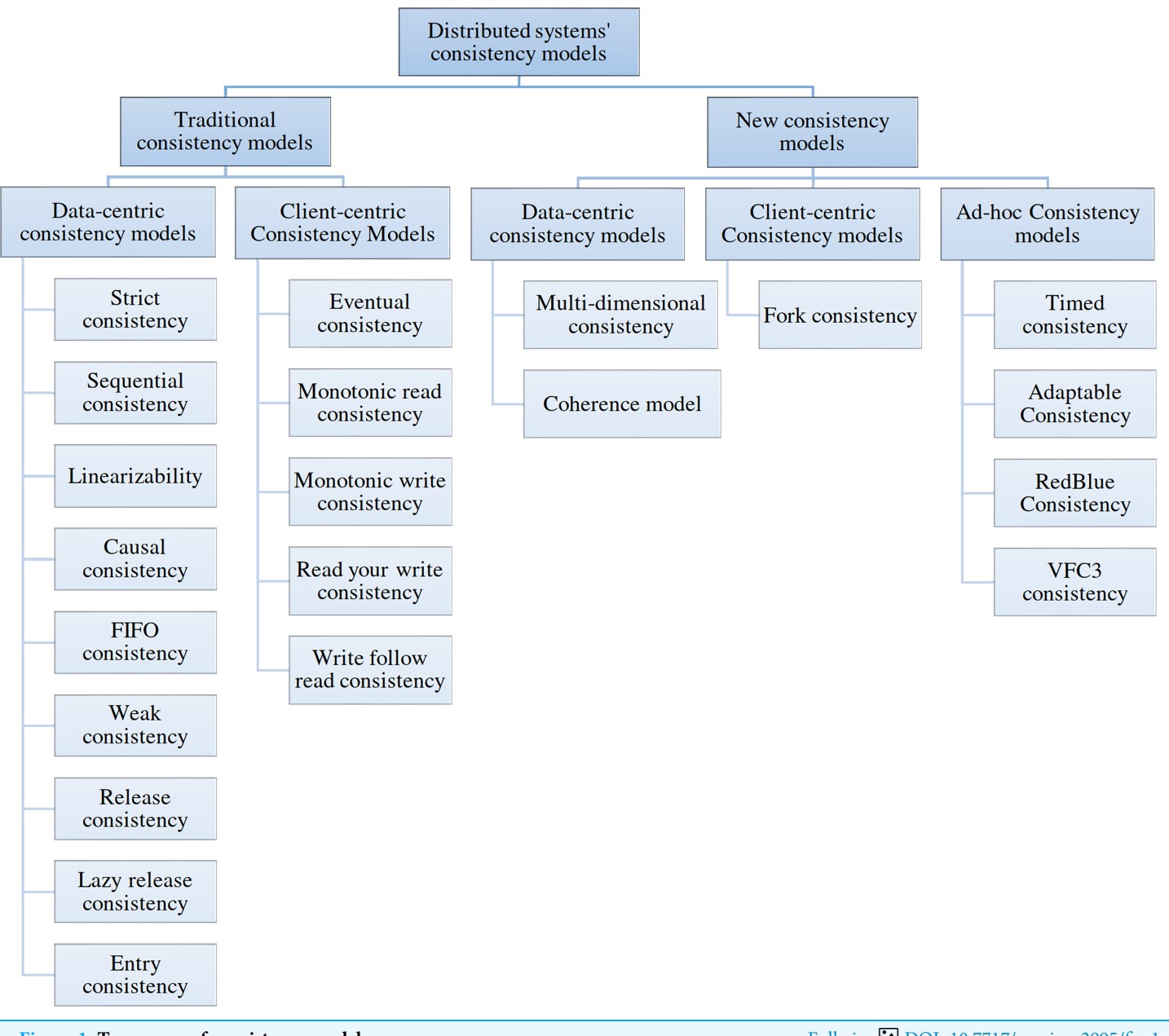

**Figure 1 Taxonomy of consistency models.**

modeling a wide range of systems such as production management and distributed systems. "Problem statement" presents the problem statement. "Definition of terms" defines the terms used in the article and provides the theoretical background. "Formal definition of colored Petri net" presents a formal definition of the colored Petri net. "Materials and Methods" presents the hierarchical proposed colored Perti net model of the causal-consistent distributed system (CCDS) by describing the color sets and functions used in the modeling, as well as color sets and functions used for state space analysis. "Discussions and Results" presents the proposed method for model checking of the developed model of CCDS and describes functions and steps used for model checking

along with three case studies. "Conclusions" provides the article's conclusion by presenting the article's contributions and the benefits and merits of the proposed model and approach. Finally, "Future works" presents future works that can be accomplished.

## RELATED WORKS

Colored Petri nets have been used for modeling job scheduling of machines in factory production lines (*Bożek, 2012*) and studying production system workflow (*Ha & Suh, 2008*; *Xiao & Ming, 2011*). Resource allocation in computerized systems aimed at deadlock avoidance using the Banker algorithm was studied by modeling using colored Petri net (*Pashazadeh, 2012*). State space analysis of the model determines whether the system's current state is secure and whether current resource allocation can cause that system's transition to another secure state.

The capability of state space analysis of Petri net models makes it applicable in the modeling and analysis of access control mechanisms in information security. Modeling a system's resources, actors, and access permission rules was investigated with the colored Petri net, and analysis of the state space of the model successfully determined conflicts of rules and redundant rules automatically (*Alamdari, Pashazadeh & Mirzamohammadzadeh, 2012*).

Modeling a take-grant protection system was accomplished by modeling resources (objects), processes, or subjects and their access using a colored Petri net (*Pashazadeh, 2013*). State space analysis of the model successfully determined possible information leakages in the system and automatically extracted scenarios for these successful attacks. Modeling the operation of a power distribution's security system and analyzing its security using probabilistic Petri net is one of the other applications of Petri net (*Ashouri & Jalilvand, 2016*).

In classical Petri net and most advanced extensions of it, such as colored Petri net, the structure of the Petri graph is static. It limits the application of Petri nets in modeling and analyzing systems with variable structures. Mobile computing systems, *ad-hoc* networks, wireless sensor/actuator networks, and service-based systems have dynamic structures because of mobility or disconnection. *Ding et al. (2022)* presented the variable Petri net (VPN) to overcome the mentioned limitation of Petri net. They introduced the concept of virtual places for abstracting interaction interfaces and described the connection and disconnection concept using new functions. VPN extends the scalability and pluggability of the Petri net. *Yang et al. (2022)* have studied the model checking of the VPN. They introduced an algorithm to convert the VPN to a Kripke structure, describing the system connection states and running states. They proposed a method to represent a property to the temporal logic formula. Using the specific connection properties of the system, they proposed optimization of the Kripke structure for targeted verification.

One of the significant challenges in using the Petri net is model checking of the unbounded Petri nets. Model checking of bounded Petri nets using its reachability graph is a common analysis method of Petri net models. However, model checking of unbounded Perti net that its coverability graph generally is modeled using the ω symbol is an active research area. *Wang et al. (2024)* presented a model-checking process of ω-independent

unbound Petri net. They proposed the generation of a new, modified, extended reachability graph and proved that it includes the complete reachability markings and transitions' sequences *via* two theorems.

Modeling specific systems like cyber-physical systems using the Petri net requires extensions to support concepts like input and output signals. *Grobelna, Wiśniewski & Wojnakowski (2019)* presented the interpreted Petri nets for modeling the control part of cyber-physical systems. *Grobelna & Szcześniak (2022)* presented the model checking of interpreted Petri nets to formally verify user-defined behavioral properties using temporal logic formulas. They write the interpreted Petri net model of the system as an abstract rule-based logical model in text format and then automatically transform it into a verifiable model in the nuXmv format. Then, they use the nuXmv model checker to verify behavioral properties using temporal logic formulas.

*Yousefi, Ghabel & Khanli (2015)* presented modeling of CC by colored Petri net. Their proposed model did not model the behavior of a CCDS. Their proposed model gives a DH of a system as input and then, based on some rules that are concluded from the meaning of the causal relationship between operations of processes, determines whether this DH is valid by a CCDS. This approach may lead to incorrect conclusions for complicated DHs and does not produce proof of their results. Their state space analysis is restricted to investigating some known effects of CCDS.

The motivation of the current article is to propose a model of a CCDS based on a known distributed algorithm. Based on the operations of input DH, the model executes a distributed algorithm while considering all possible delays in message deliveries based on the predefined constraints. By simulation run of the model, all possible scenarios under a CCDS are generated. Finally, the validation of the given DH is proved by an automatic state space analysis of the model. In the case of validity, the complete scenario that produces a given DH is reported as the validation proof.

The size of the proposed model's state space graph (SSG) increases exponentially by increasing the number of processes and their operations. Three effective methods are presented in this article against the state space explosion of the model. Producing validation proof of a given DH requires investigating the proposed model's big SSG using automated tools such as CPN tools and developing the required state space analysis functions, which are accomplished in this article. The following article sections present a complete description of the proposed model.

## PROBLEM STATEMENT

However, the main problem is that the concepts of various consistencies are new and unfamiliar to graduate students. Learning these consistencies is difficult and time-consuming, and they face new problems. Distributed programs under these types of consistencies behave in a way that looks strange to students and reduces their desire to use them and develop the system under their conditions.

CC is an easy and more understandable consistency model than other consistency models. This article aims to model one of the well-known, simple, and widely used algorithms for implementing CC in DSs presented in most textbooks on advanced

operating systems or distributed systems at the graduate level. This article aims to model the behavior of a DS with the support of CC under the specified algorithm, which implements the CC using Fideg's vector clocks (*Tanenbaum & Steen, 2006*; *Coulouris et al., 2011*; *Kshemkalyani & Singhal, 2011*). However, modeling using one of the formal methods is intended so that we can prove the validity of our answers. So, we have used colored Petri nets as a formal modeling tool.

Now the question is, what features do we want to prove? This article aims to give a distributed history (DH) of a DS's behaviors as input of the DS's model, and as the output, the system automatically checks whether a CCDS can produce such a DH. If DH is valid (possible), state space analysis of the model provides a scenario of DS's behavior as proof of the answer that can create input DH. We prove the validity of the output answer by generating the proposed model's state space graph (SSG) for the given input and model checking of it. The proposed model allows users to check whether the input DH is a valid picture of a CCDS. It is similar to typical exam questions of the distributed systems or modern operating system course. The model automatically finds the answer in a very short time. If yes, it provides the simplest (shortest) proof scenario.

In many cases, the input sample DH may be so complex that individual students' answers are different, causing discussions between them. This model can be used as a learning tool to generate the proof scenario automatically. The proof scenario is a sequence of DS's events that produces the input DH.

Another application of this modeling is to prepare a basis for studying existing distributed algorithms or developing new ones for enforcing a consistency model in a DS. Developing a formal model of an algorithm allows us to automatically check and detect the hidden bugs of the distributed algorithm. Usually, the developers of a distributed algorithm try to prove the correctness of the algorithm after initial tests and ensure that the algorithm works correctly. The Petri net model of an algorithm is usually unable to prove the correctness of a distributed algorithm. However, if the algorithm has hidden bugs, it can help find and extract failure scenarios automatically. The proposed model examines all the behaviors that the DS can provide under the selected consistency algorithm, and if the system can generate the input DH, it can extract all the scenarios that can lead to that history. The proposed model returns the shortest proof scenario *via* model checking of the SSG. The correctness of the distributed algorithm we used for enforcing CC has been presented before and is not our concern in this article.

# DEFINITION OF TERMS

This part presents all required background definitions, concepts, and algorithms used in this article.

## Abstract model of DS

This article considers a simple abstract model of a DS and models it using the colored Petri net. Let us assume that our DS consists of a few systems, and a process runs on each. From now on, we use the term process to represent a system and the process that runs on it. Let us assume that DS comprises a group $g$ of independent processes $P_i$ ($i = 1, 2\ldots, N$), and a

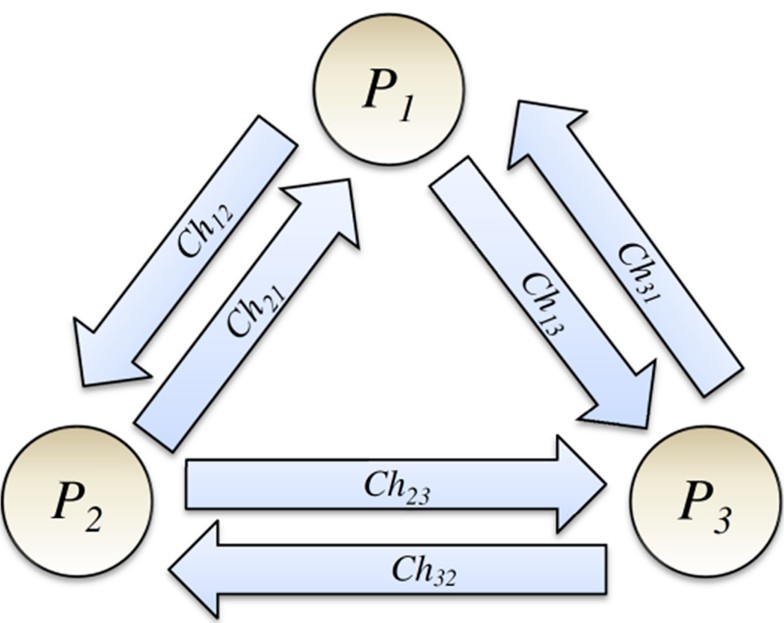

**Figure 2 Processes and their communication channels.**

communication channel exists between each pair of processes, as shown in Fig. 2 (*Tanenbaum & Steen, 2006*; *Coulouris et al., 2011*). The communication channel is a usual abstract concept in DSs and is assumed unidirectional. Channel $Ch_{ij}$ is used to send messages of the process $P_i$ to process $P_j$. Based on the definition of FIFO-ordered deliver property for communication channels (also known as FIFO-ordered channels), If process $P_i$ sends two messages $m_1$ and then $m_2$ to process $P_j$ (*via* channel $Ch_{ij}$), message $m_1$ will be delivered before message $m_2$.

*Assumption 1*: Let us assume that all communication channels have FIFO-ordered delivery property.

Let us assume that communication channels are reliable and no message will be lost, but the message delivery delay is not deterministic. Some algorithms for implementing reliable multicast have been developed (*Coulouris et al., 2011*). In a DS, when a process $P_i$ writes value $v$ on a replicated variable $x$, the write message is multicasted to replication group $g$ of processes. If process $P_i$ reads the value of the replicated variable $x$, no message multicasting is required. In this article, only a single group of processes is considered.

In a DS, each process has its local copy of shared variables. When a process $P_i$ in a group $g$ of processes writes value $r$ on data item $x$, it multicasts this *via* sending messages to the group members in DS to notify them. $Multicast_{P_i}(g, m)$ denotes the multicast of message $m$ to the member of the group $g$ by process $P_i$. When the sender process is clear, for simplicity, we denote $Multicast(g, m)$.

## Concept of causal consistency

This part presents the concept and formal definition of CC. First, the concept of the happens-before relationship between two events is presented. Then, the concept of Fideg's

clocks and their properties are introduced. Then, the concept of CC and its formal definition are presented.

### Happens-before relationship

Happens-before is a logical, ordering relationship between two events. If event $a$ happens before an event $b$, we denote it as $a \rightarrow b$. When both events happen in the same process, $P_i$, the notation $a \rightarrow_i b$ is also used. Happens-before relationship has the following properties (*Tanenbaum & Steen, 2006*; *Coulouris et al., 2011*):

1. If two events, $a$ and $b$, happen in the same process and event $a$ occurs before $b$, then $a \rightarrow b$.
2. If $a$ is the event of sensing a message $m$ (send event) and $b$ is the event of receiving that message (receive event), then $a \rightarrow b$.
3. If $a \rightarrow b$ and $b \rightarrow c$, then $a \rightarrow c$ (transitivity property).

An event that is neither a send nor receive event is called an internal event.

*Assumption 2*: Let us assume that never two events occur simultaneously in a process.

A happens-before relationship defines partial ordering between events of a distributed system. If $a \nrightarrow b$ and $b \nrightarrow a$, then $a$ and $b$ are concurrent, and we denote $a \parallel b$.

### Fidge clocks

For each process $P_i$ ($i = 1, 2\ldots, N$) in the group $g$, we consider vector clocks (vector timestamps) $V_i^g$, which is a vector of whole numbers whose dimension ($N$) equals the number of processes in the group. Sometimes, we omit the group name's superscript of a vector for simplicity. $V_i[j]$ denotes $j^{\text{th}}$ element from vector clocks of process $P_i$. Fidge clocks follow these rules (*Tanenbaum & Steen, 2006*; *Coulouris et al., 2011*):

1. Vector clocks of all processes are initialized as follows:
   $V_i[j] = 0, \forall\, i, j = 1, 2\ldots, \text{N}$
2. When an internal or send event happens in a process $P_i$, the timestamp $V_i[i]$ will be incremented by one. It enforces the happens-before relationship between the previous and the current events in the process $V_i$.
3. When a process $P_i$ sends a message $m$ to process $P_j$, process $P_i$ first updates its vector timestamps based on rule 2, then sends it along with message $m$ to the process $P_j$.
4. When a process $P_j$ receives a message $m$ from process $P_i$ ($i \neq j$), it updates its vector timestamps based on the received timestamps vectors as follows:

   4.1. Increments $j^{\text{th}}$ element of its vector timestamps by one: $V'_j[j] = V_j[j] + 1$, $V'_j$ denotes updated vector timestamps of process $P_j$.

   4.2. Assign the maximum value of $k^{\text{th}}$ element of vector timestamps of process $P_j$ and received message's vector timestamps (sender process's vector timestamps) for $k \neq j$: $\forall\, k \in \{1, 2.., \text{N}\}$, $k \neq j$, $V'_j[k] = \text{maximum}\,(V_j[k], V_m[k])$, $V_m$ denotes vector timestamps of the incoming message.

$V_j[j]$ is the knowledge of process $P_j$ about the events that have happened in it (the number of events that occurred at $P_j$). $V_j[k]$ represents the number of events that process $P_j$ knows that have happened in process $P_k$.

### Fidge clocks' properties

A strong property of Fidge's clocks is that if $a \rightarrow b \Leftrightarrow V(a) < V(b)$. $V(a)$ denotes the vector timestamps of event $a$. Let us define the relationships between two timestamp vectors (*Tanenbaum & Steen, 2006*).

$V(a) = V(b)$ if $\forall\, k \in \{1, .., N\}, V(a)[k] = V(b)[k]$
$V(a) \neq V(b)$ if $\exists\, k \in \{1, .., N\}, V(a)[k] \neq V(b)[k]$
$V(a) \leq V(b)$ if $\forall\, k \in \{1, .., N\}, V(a)[k] \leq V(b)[k]$
$V(a) < V(b)$ if $V(a) \leq V(b)$ and $V(a)[k] \neq V(b)[k]$.

Implementing the Fidge's clocks can be accomplished by applying the mentioned rules on the processes of a DS.

***Note***: operation, implementation and properties of Fidge's clocks are independent of assumption 1.

### The concept of the causal relationship of events

Let us assume that in a discussion group, a process $P_i$ writes a message with a typo fault and then writes the next message to correct it. These two writes are causally related to each other. Let us assume that process $P_i$ writes value $r$ on variable $x$, and then process $P_j$ knows this (*via* delivery of multicast write operation message of process $P_i$) and writes value $q$ on variable $y$. The writing of $P_j$ is potentially causally related to the writing of process $P_i$. Maybe the writing of the process $P_i$ has influenced the writing of $P_j$. If process $P_j$ writes a value on variable $y$ before knowing the write of process $P_i$ (write message delivery of $P_i$) and process $P_i$ writes a value on variable $x$ before knowing the write operation of process $P_j$, these two events are considered concurrent and do not have a causal relationship.

### The causal relationship of messages

We pass from the events and concentrate on multicasting messages. From now on, any two messages with the happens-before relationship are considered causally related. We defined the happens-before relationship for events. What is the happens-before relationship between messages? We consider the happens-before relationship between sending events of messages as the happens-before relationship between messages. We mentioned that only the write operations of a process on a replicated variable will be multicasted as a message to the group's members. Now, we can introduce the formal definition of causal ordered message delivery and the CC model.

### Formal definition of the CC

The definition of casual ordering of messages is as follows:

If multicast $(g, m) \rightarrow$ multicast $(g, m')$, then any correct process in group $g$ that delivers $m'$, will deliver $m$ before $m'$ (*Coulouris et al., 2011*).

In other words: $m \rightarrow m' \Rightarrow \forall\, p \in g$: $deliver_p(m) \rightarrow deliver_p(m')$

---

**Algorithm 1** Pseudo code of causal ordering using vector timestamps.

1:  //  Algorithm for group g's members $P_i$ $(i = 1, 2 \ldots, N)$
2:  Initialization of the Fidge's vector clocks of all group g's members
3:  $V_i^g[j] := 0$   $(i, j = 1, 2 \ldots, N)$
4:  CO-multicast of message $m$ to g's members by process $P_i$:
5:              1. $V_i^g[i] := V_i^g[i] + 1$
6:              2. B-multicast ($<V_i^g, m>$)
7:  When $P_j$ B-delivers ($<V_i^g, m>$) from $P_i$ $(j \neq i)$, with $g = $ group($m$):
8:              1. It places $<V_i^g, m>$ in its hold-back queue
9:              2. Wait until $V_i^g[i] = V_j^g[i] + 1$ and $V_i^g[k] \leq V_j^g[k](k \neq i)$;
10:             3. CO-deliver m; // after removing $<V_i^g, m>$ from the hold-back queue
11:             4. $V_j^g[i] := V_j^g[i] + 1$;

---

$deliver_p(m)$ means delivery of message $m$ by process $p$ using an algorithm that guarantees CC. Causal consistent DS guarantees that all processes see the causally related messages (writes) in the correct order. Concurrent writes can be seen in a different order by processes (*Tanenbaum & Steen, 2006*). CC means that if two events have a causal relationship, all other DS processes correctly understand this relationship.

## Algorithms for implementing CC in DS

Many researchers have studied the implementation of CC in DS. *Birman & Joseph (1987)* introduced the concept of causal ordering of messages. They presented the algorithm for implementing causal ordering of messages in DS at Cornell University in the ISIS system (*Birman, Joseph & Schmuck, 1987*). Their proposed method has the overhead of sending many messages. *Schiper, Eggli & Sandoz (1989)* presented an algorithm for implementing CC in DS that needs the transmission of fewer messages than the previous method. Their proposed method needs the attachment of the vector timestamp of the message's sending event along with the content of the message. Their proposed method needs each process to keep a buffer named the ORD-Buff and send it *via* message to other processes. *Raynal, Schiper & Toueg (1991)* presented an algorithm for implementing CC requiring sending a two-dimensional matrix with each message. Each process has a vector and a two-dimensional matrix as local variables used by their proposed algorithm. *Kshemkalyani & Singhal (1996, 1998)* presented their proposed optimal space and time algorithm.

*Birman, Schiper & Stephenson (1991)* presented an algorithm for implementing the CC in DS taught in the textbook by *Coulouris et al. (2011)*. We modeled this algorithm in the current article. This algorithm requires Fidge's vector clock for logical time synchronization of processes and needs the concept of hold-back and delivery queues. The algorithm for implementing CC in DS is presented in Algorithm 1.

When a process $P_i$ multicasts a message $m$ to all other members of group $g$, it updates its vector clocks as $V_i[i] = V_i[i] + 1$ and then multicast message $<V_i, m>$. When a process $P_j$ delivers a multicast message from the sender process $P_i$, the message is placed in the

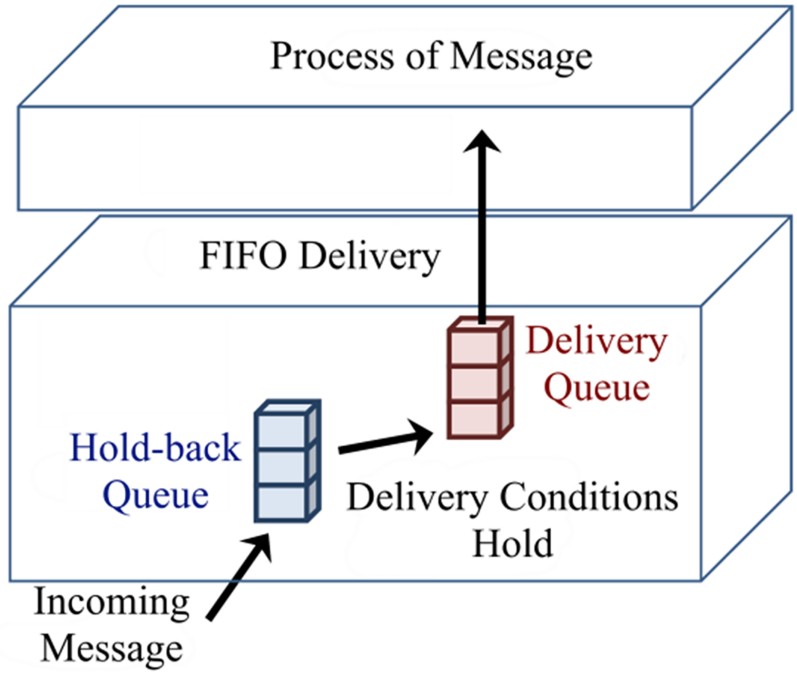

**Figure 3 Hold-back and delivery queues of a process.**

hold-back queue. Every message in the hold-back queue that meets the following conditions will be moved to the delivery queue.

$$\forall\, k \in \{1, .., N\}, k \neq i, V_j[k] \geq V_i[k].$$

Note: the condition $V_i^g[i] = V_j^g[i] + 1$ in line 9 of the algorithm is not required if the communication channels support the FIFO-ordered delivery. This article assumes all channels have FIFO-delivery property *via* implementing one of the known algorithms for this abstraction.

Figure 3 shows delivery and hold-back queues. When a message enters the delivery queue, vector clocks of process $P_j$ will be updated as $V_j[j] = V_j[j]+1$, which can be considered as the delivery of a message. Process $P_j$ completely updates its vector clocks based on the Fidge algorithm when processing the next message from the delivery queue (*Coulouris et al., 2011*).

Figure 4 shows a DH of a CCDS. When the issued message $m'$ of process $P_1$ receives to process $P_3$, it will be inserted in the hold-back queue of process $P_3$ and stay there until it meets the required condition for transferring to the delivery queue of process $P_3$. This condition will be met when the issued message $m$ by process $P_2$ is delivered to process $P_3$. When message $m'$ is received by process $P_3$, the vector clock of the message represents that some event happened in process $P_2$, but process $P_3$ is unaware of this event happening in $P_2$. Therefore, process $P_3$ postpones the delivery of message $m'$ issued by process $P_1$ until it delivers message $m$ from process $P_2$.

In this article, a formal colored Petri net model of a CCDS that is composed of three processes is presented. The proposed hierarchical model was designed in two abstraction

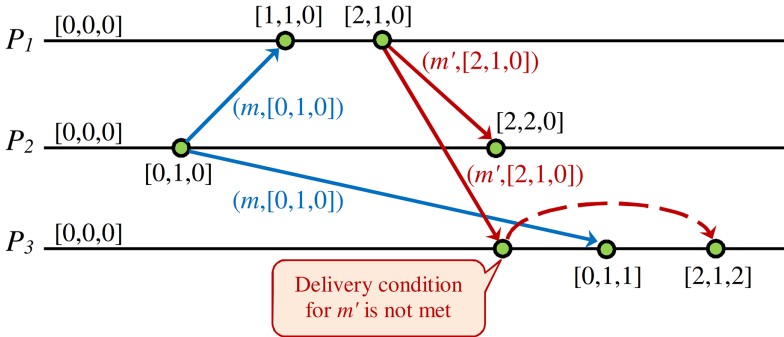

**Figure 4 Example of delay in processing received message to guarantee CC in DS.**

**Table 1 DH of case study 1.**

| Process | Global sequence number | | | | |
|---|---|---|---|---|---|
| | 1 | 2 | 3 | 4 | 5 |
| $P_1$ | $W(x){:}1$ | | | | |
| $P_2$ | | $R(x){:}1$ | $W(y){:}2$ | | |
| $P_3$ | | | | $R(y){:}2$ | $R(x){:}0$ |

layers. A complete description of the proposed model is presented in the following sections. Let us assume that time synchronization of all replicas is performed by logical Fidge clocks (vector clocks). The proposed model's input is a sample DH of a CCDS.

***Case study 1.*** Table 1 shows a sample history of operations in a CCDS with three processes. The global sequence number represents the partial order of operations in DS. This article assumes that all state variables' initial value is zero. The heading of the table's rows represents the names of processes, and rows contain the operations of each process. At the first row of the table, $W(x){:}1$ denotes that process $P_1$ writes value one on the replicated data-item $x$ at global sequence order 1.

   The write operation of process $P_2$ ($W(y){:}2$) has a causal relationship with the write operation of process $P_1$ ($W(x){:}1$). Let us consider the worst-case scenario in which process $P_2$ has instruction $y = x * 2$. In this case, the written value of the variable $y$ has been influenced by the read value of the variable $x$. Process $P_3$ does not understand this relationship correctly. The first read of $P_3$ ($R(y){:}2$) sees value two, and the second read of this process ($R(x){:}0$) sees value zero, then we are faced with a DS that does not support CC.

   Petri net is a formal method that is designed to model concurrent systems. Various extensions of the classical Petri net have been proposed. One of the most successful is the colored Petri net, which can model synchronous and asynchronous communications (*Jensen & Kristensen, 2009*). The modeling power of colored Petri net is extensively extended by (1) using data type (color set) for tokens and (2) using machine learning (ML) artificial language that is a formal language based on the Lambda calculus that preserved the formal property of Petri net (*Paulson, 1996*). One of the best tools for modeling by colored Petri net is the CPN tool used in this article (*Beaudouin-Lafon et al., 2001*).

# FORMAL DEFINITION OF COLORED PETRI NET

This part presents the formal definition of a non-hierarchical colored Petri net and colored Petri net module (*Birman, Joseph & Schmuck, 1987*).

**Definition 1.** A *non-hierarchical colored Petri net* is a nine-tuple $CPN = (\Sigma, P, T, A, V, C, G, E, I)$ in which:

(1) $\Sigma$ is a finite set of non-empty types, called *color sets*,

(2) $P$ is a finite set of *places*,

(3) $T$ is a finite set of *transitions* such that $P \cap T = \varnothing$,

(4) $A \subseteq (P \times T) \cup (T \times P)$ is a finite set of directed *arcs* in which $P \cap A = T \cap A = \varnothing$,

(5) $V$ is a finite set of *typed variables* such that $\forall v \in V : Type[v] \in \Sigma$,

(6) $C: P \to \Sigma$ is a total *color set function* that assigns a color set to each place,

(7) $G: T \to EXPR_v$ is a *guard function* that is defined from $T$ into expressions such that $\forall t \in T : Type[G(t)] = Bool$,

(8) $E: A \to EXPR_v$ is an *arc expression function* that assigns an arc expression to each arc $a$: $\forall a \in A : Type[E(a)] = C(p)_{MS}$ where $p$ is the connected place to the arc $a$,

(9) $I: P \to EXPR_0$ is an *initialization function* that assigns an initial expression to each place $p: \forall p \in P : Type[I(p)] = C(p)_{MS}$ (*Birman, Joseph & Schmuck, 1987*).

**Definition 2.** The following concepts are defined for a colored Petri net $CPN = (\Sigma, P, T, A, V, C, G, E, I)$:

(1) $M: P \to C(p)_{MS}$ is *marking* (function) that maps each place $p \in P$ into a multiset of tokens $M(p) \in C(p)_{MS}$.

(2) The *initial marking $M_0$* is defined by $\forall p \in P : M_0(p) = I(p)$.

(3) To each transition $t$, $Var(t)$ denotes the *variables of transition $t$*, $Var(t) \subseteq V$, and consists of the free variables in the guard of $t$ and the arc expressions of arcs connected to $t$.

(4) A *binding* of a transition $t$ is a function $b$ that maps each variable $v \in Var(t)$ into a value $b(v) \in type [v]$. $B(t)$ denotes the set of all bindings of transition $t$.

(5) A pair $(t, b)$ such that $t \in T$ and $b \in B(t)$ denotes *a binding element*. $BE(t)$ denotes the set of all binding elements of transition $t$ and is defined by $BE(t) = \{(t, b) \mid b \in B(t)\}$. $BE$ denotes the set of all binding elements in a CPN model.

(6) A *step* $S \in BE_{MS}$ is a non-empty, finite multiset of binding elements.

**Definition 3.** A *colored Petri net module* is a four-tuple $CPN_M = (CPN, T_{sub}, P_{port}, PT)$, where:

(1) $CPN = (\Sigma, P, T, A, V, C, G, E, I)$ *is a* non-hierarchical colored Petri net,

(2) $T_{sub} \subseteq T$ is a set of *substitution transitions*,

(3) $P_{port} \subseteq P$ is a set of *port places*,

(4) $PT : P_{port} \rightarrow$ {*IN, OUT, I/O*} is a *port type function* that assigns a port type to each port place.

## MATERIALS AND METHODS

This part of the article presents our method for modeling a CCDS using colored Petri net and analyzing its state space graph for model-checking our intended properties. This part presents the proposed hierarchical colored Petri net model of CCDS. First, color sets used in the model are introduced, and then the proposed model's top and second levels are described. All places and transitions of the model are presented along with functions developed in the model.

### Color sets of the model

The proposed colored Petri net model in this article is parametric. Two important parameters are the number of constituting processes in the distributed system's model and the number of data items that processes update and read them. The following two constants determine the number of processes and the number of variables in the model.

val NoProcs = 3;

val NoVars = 2;

Definitions of color sets that are used in the model and represent data types of places and tokens are as follows:

colset PROCESS = index proc with 1..NoProcs;

colset CLOCKSVECTOR = list INT;

colset STATEVARIABLE = record varName: STRING * value: INT;

colset LOCALSTATE = list STATEVARIABLE;

colset OPERATIONTYPE = with R | W;

colset OPERATION = record process: STRING * opr: OPERATIONTYPE * varName: STRING * value:INT * gSeqNo:INT;

colset OPERATIONS = list OPERATION;

colset MESSAGE = record varName: STRING * value:INT * vector: CLOCKSVECTOR * sender: PROCESS;

colset MESSAGELIST = list MESSAGE;

colset SENDMESSAGE = record process: PROCESS * msgs: MESSAGELIST;

Color set PROCESS is of type index and is defined to represent a process identity. Color set CLOCKSVECTOR is a list of integer numbers used to represent a process's Fidge vector clocks. Color set STATEVARIBLE is a record with fields: (1) *varName* that is of type string to represent the name of a variable and (2) *value* that is of type int and represents the value of a variable. Color set LOCALSTATE is a list of type STATEVARIABLE and is defined for representing values of variables of a process that may be replicated between processes. Color set OPERATIONTYPE is an enumerated type defined to denote read and write operations on variables. Color set OPERATION is a record with five fields. The first field with the name *process* represents the name of the process that will execute this operation.

The second field, *opr*, represents the type of operation (read/write). The third field, named *varName*, represents the variable name on which the operation will be performed. The fourth field with name *value* represents the variable's value in case of read operation or value that must be stored in the variable in case of write operation. The last field, named *gSeqNo*, represents this operation's global sequence number. Color set OPERATIONS is a list of type OPERATION used to represent process operations.

Operations on the state variables of a process will be stored according to the ascending value of the global state number of operations. Color set MESSAGE is a record with four fields: the first field, named *varName*, represents the variable's name. The second field, named *value*, represents the variable's value, the third field, named vector, represents the Fidge vector clocks of the message issuer process, and the last field, named sender, represents the message issuer process. Color set MESSAGELIST is a list of types of MESSAGE. Color set SENDMESSAGE is a record with two fields. The first field, named *process* represents the message issuer process, and the second field, named *msgs* is the list of issued messages by this process.

## Proposed colored Petri net model of system

This article's proposed CCDS model is hierarchical in two abstraction layers. The top-level model of the proposed system model with initial markings of case study 2 is shown in Fig. 5.

Operations of process $P_i$ is stored in place OperationsPi with color set OPERATIONS and values of its state variables is stored in place LocalMemoryPi with color set LOCALSTATE and process identity is stored in place Pri with color set PROCESS. Place GlobalStep with color set int represents the global sequence number of the system with initial value one. Place outbox with color set SENDMESSAGE represents issued messages of processes and place inBox with color set INCOMEMESSAGE represents incoming messages of all processes that will be stored in their hold-back queues. Place inBox in the top-level model of Fig. 5 plays the rule of the hold-back queue for all processes. The marking of this place is a list in which each element represents a list of messages stored in the hold-back queue of each process in CCDS.

Transition StepUpdate is responsible for increasing the global sequence number of the model. The transition network plays the role of communication networks and channels in DS. Substitution transition Ri represents operations of process Pi, an instance of the Replica module. Colored Petri net model of the Replica module of substitution transition of process $P_2$ is shown in Fig. 6. Place LocalMemory is an I/O port related to the place LocalMemoryPi of process $P_i$ in the top-level model and is related to socket OperationsPi of process $P_i$.

I/O port OutBox is related to socket outBox, I/O port InBox is related to socket inBox, and I/O port StepNumber is related to socket GlobalStep in the top-level model of Fig. 5. I/O port ProcessNumber is related to the place Pri of process $P_i$ in the top-level model of the system. Place Clocks with color set CLOCKSVECTOR is an internal place and represents Fidge vector clocks of a process. Transition ExecuteOperation picks up an operation from the beginning of the operations list of a process on each fire and executes it.

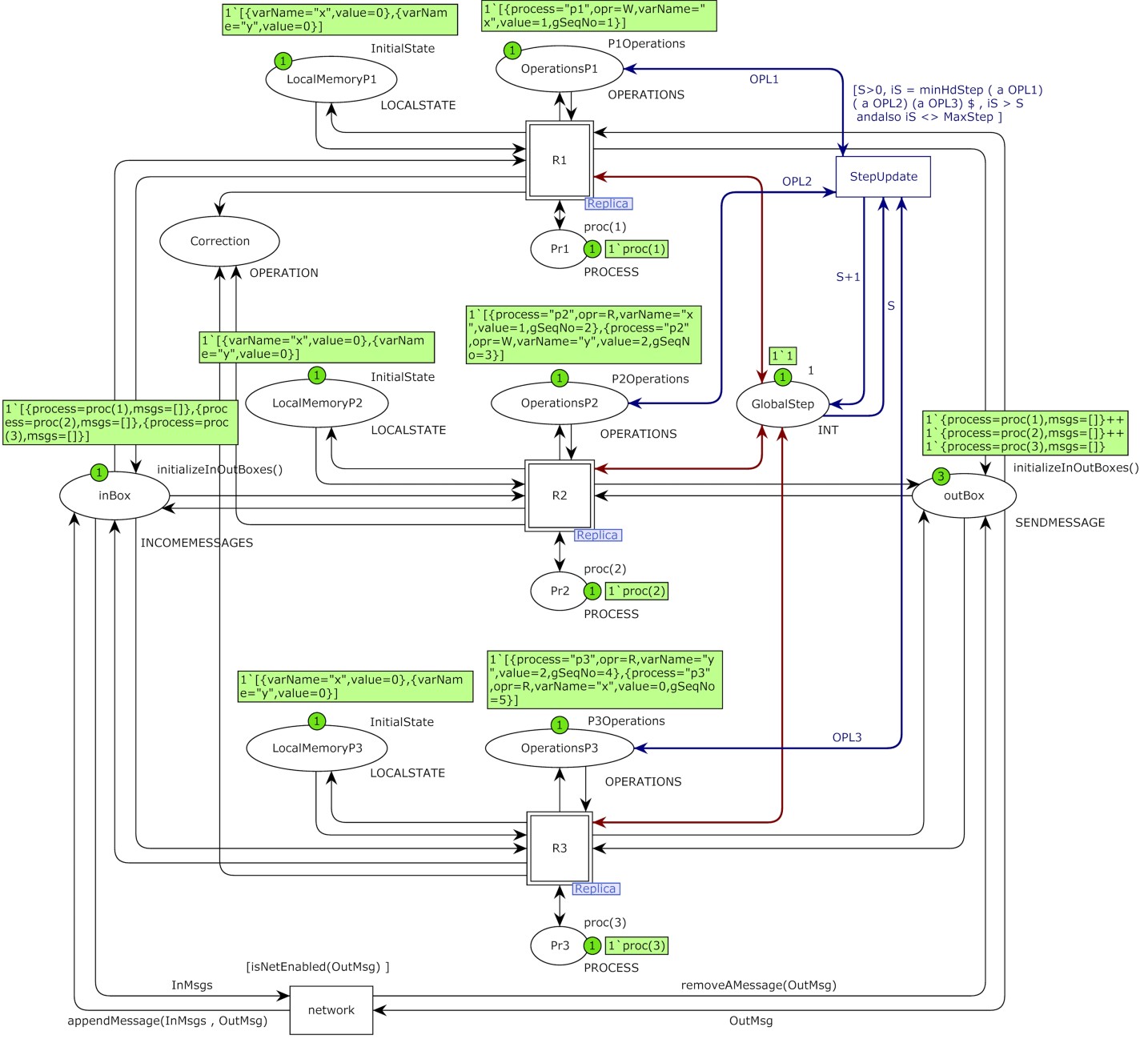

**Figure 5  Top-level of the proposed model for CCDS with three constituting processes with the initial marking of case study 1.**

Transition PickupMessage picks up a message from one of the channels in each firing. Initial markings of Figs. 5 and 6 are based on the DH of Table 1. Operations of process $P_2$ are defined using constant P2Operations based on Table 1, and they are used as initial marking of place OperationsP2 in the model of Fig. 5.

  val P2Operations = [{process="p2", opr=R, varName="x", value=1, gSeqNo=2}, {process="p2", opr=W, varName ="x", value=2, gSeqNo=3}]

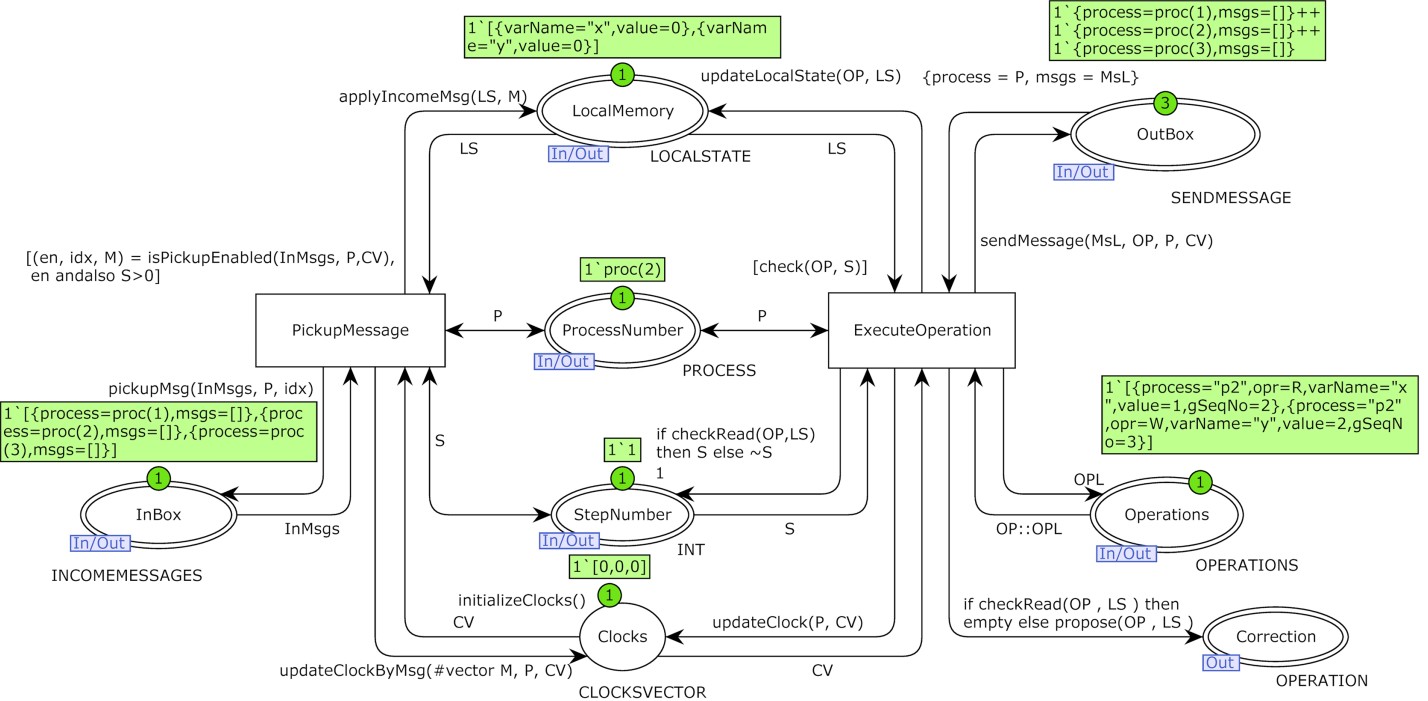

**Figure 6** Model of replica module of process P₂ with the initial marking of case study 1.

The proposed model has two important parameters, NoVars and NoProcs. NoVars determines the number of variables (data items) of the input DH and the model. NoProcs determines the number of processes in the input DH and the model. These variables are used in the model's functions to increase scalability.

## Functions of the proposed model

This part of the article presents the functions developed and used in the proposed model. Figure 7 displays the structure chart of the proposed model's functions. The system's top-level model uses nine functions displayed in the figure's top left corner.

The signature and description of all nine functions used in the proposed model's top-level module are presented in Table 2. Row 5 to 9 in Table 2 provide the fold technique that permits the variable number of input parameters of functions that standard ML and CPN ML do not support.

Function *appendMessage* is used in the output arc of the transition network in the top-level model of the proposed system. The function *stepUpEn* is used as a guard condition for transition StepUpdate. Global sequence number zero represents the unpermitted state of the system. It denotes a true deadlock of the model and disables all transitions. Value zero will be generated when a process executes a read operation based on the history, and the value of the local memory of a process is not the same as the expected value that is indicated in history. It means that the history of DS is not reachable by CCDS, proving that the history is not valid. Therefore, after the proof is completed, the execution of the model will be stopped to overcome the state space explosion problem.

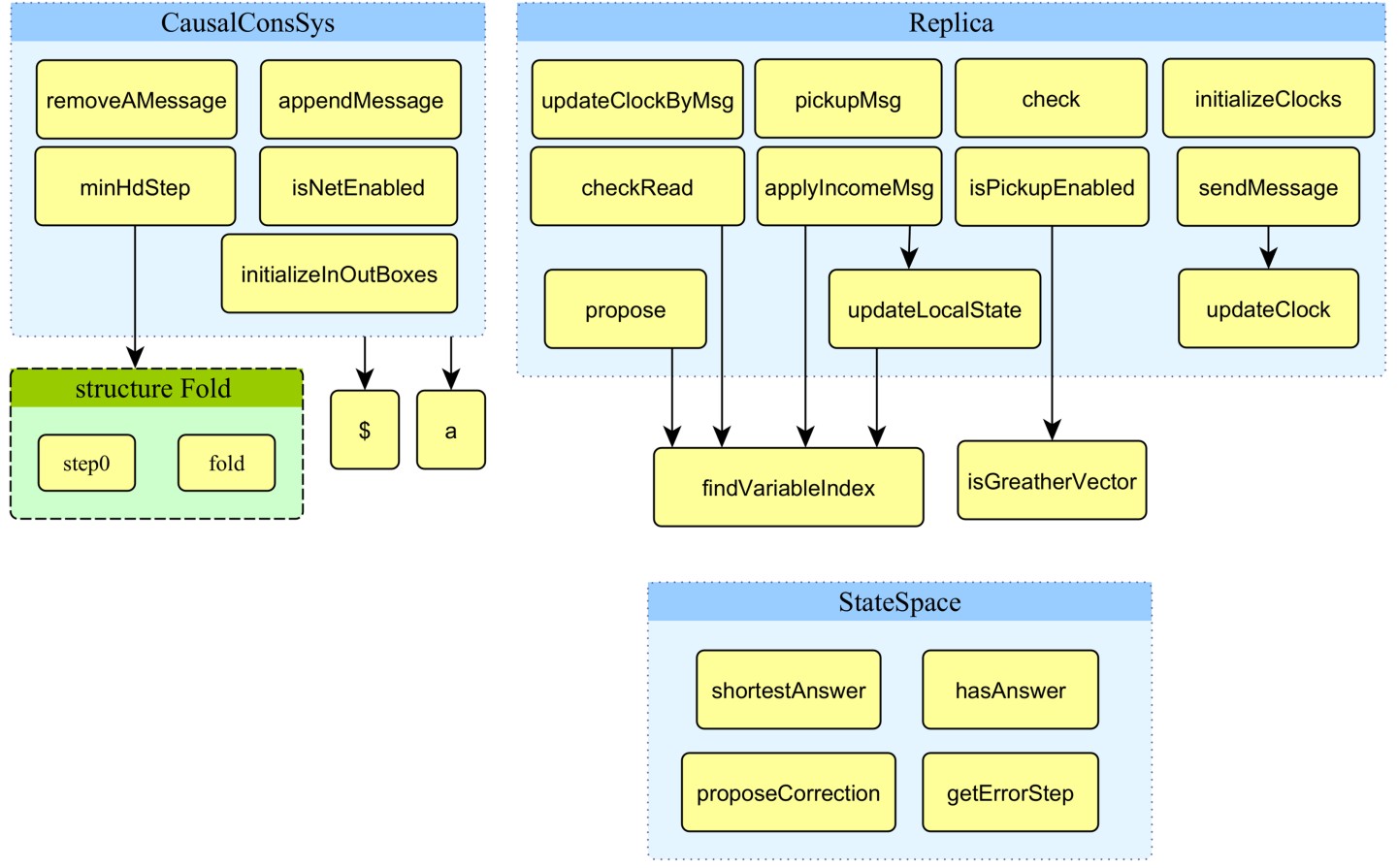

**Figure 7** Structure chart of functions of the proposed model.

**Table 2 Signature and description of functions used at the prime module of the proposed model.**

| Index | Function signature | Description |
|---|---|---|
| 1 | fun appendMessage(L: INCOMEMESSAGES, m: SENDMESSAGE): INCOMEMESSAGES | The function gets (1) a list of incoming messages to the hold-back queue of all processes and (2) a new incoming message as input parameters and returns the updated list of incoming messages to hold-back queues. It appends incoming messages to the end of hold-back queues of all other processes in the network. If the msgs field of the incoming message is empty or the number of elements in the hold-back list is less than the number of total processes of the system, then the function does not update the list. If process P1 issues the message, then this message will be added to the end of hold-back queues of all other processes. |
| 2 | fun isNetEnabled(m: SENDMESSAGE): BOOL | The function gets a record containing the list of messages in the outbox of a process as input parameters and returns the true value if the process has any message that must be sent. The function returns false when no message exists to send. |
| 3 | fun removeAMessage(m: SENDMESSAGE): SENDMESSAGE | The function gets a record containing the list of a process's messages that must be sent. This function will be called when the process has any message to send. The function removes the first message from the list and returns the list of remaining messages in the record as the result. |
| 4 | fun initializeInOutBoxes(): : INCOMEMESSAGES | The function returns a list. This list shows that no incoming/outgoing messages to/from all processes existed initially. |
| 5 | fun $ (a, f) | Calls function f with parameter a. |
| 6 | fun a(i: OPERATIONS) | Calls Fold.step0 |

| Index | Function signature | Description |
|---|---|---|
| | **Table 2 (continued)** | |
| 7 | fun fold (a, f) g | Calls function g with parameters a and f, respectively. |
| 8 | fun step0 h (a, f) | Calls fold (h a, f). |
| 9 | fun minHdStep Z | This function uses the method fold of the structure Fold to accept the variable number of parameters. The input parameters of this function are, respectively, a list of remaining operations of processes. Operations of the processes in the input DH have been sorted with the increasing field of the gSeqNo. The function looks at the head of each process's operations and returns the minimum value of gSeqNo filed of them. If all processes have been executed, the list of each process's operations will be empty, and the function will return the constant MaxStep. |

The proposed model assumed that the maximum number of operations of each process is limited to up to 100 operations, and operations of each process are stored based on the ascending order of their gSeqNo field of operations that represents the global sequence number of each operation. If the value of the global sequence number is zero or all operations of all processes have been executed (therefore, the list of their operations is empty), function *stepUpEn* returns the false value as a result. As far as a process has an unexecuted operation in that its sequence number is equal to the global sequence number of the system, the global sequence number of the model will not increase. It means that as long as a process has an unexecuted operation with a low sequence number, an operation with a higher sequence number of no process can execute. The initial value of the global sequence number of the model is one, and operations of all processes start with a sequence number one.

The system's second-level module of the model (replica module) uses 13 functions displayed on the right side of Fig. 7. The signature and description of all 13 functions used in the proposed model's top-level module are presented in Table 3. The arrow from one function to another represents it calls the function located in the destination of the arrow. Two functions in the low right corner of Fig. 7 are not directly used in the module replica.

The function *checkRead* is used as a guard condition of transition ExecuteOperation in the low level of the proposed model in Fig. 6. Based on the distributed protocol of implementing CC, if items of vector clocks of the issuer process of the message (except the item that represents the index of issuer process) are greater than (be fresher than) items of vector clocks of delivering process, it means that information of delivering process about other processes beyond the issuer one is not up-to-date. It means there is at least a message that the delivery process is unaware of. In this case, the function returns the false Boolean value. The *isPickupEnabled* function is used as a guard condition of transition pickupMessage in the low-level model of Fig. 6. This function searches the hold-back queue of a process and, if it finds any message that meets the delivery condition, returns (1) the true Boolean value, (2) the index of the message in the delivery queue, and (3) that message as a result. The function *pickupMsg* is used on the output arc of transition pickupMessage that is connected to place InBox in the low-level model of Fig. 6. After enabling this transition by calling the guard function *isPickupEnabeled*.

**Table 3 Signature and description of functions used at the replica module of the proposed model.**

| Index | Function signature | Description |
|---|---|---|
| 1 | fun findVariableIndex(LS: LOCALSTATE, vName: STRING): INT | The function gets the local state of a process containing a list of state variable values as the first input parameter. It gets the name of a state variable as the second input parameter and returns the value of the variable of type integer as the output result. |
| 2 | fun updateLocalState(OP: OPERATION, LS: LOCALSTATE): LOCALSTATE | The function gets an operation as the first input parameter and the state of a process as the second input parameter and returns the updated status of the process after execution of the input operation. The state of local variables of the process will not change in case of read operation. |
| 3 | fun checkRead(OP: OPERATION, LS: LOCALSTATE): BOOL | The function gets an operation as the first input parameter and the local state of a process as the second input parameter. This function finds the value of a variable *via* a search in the list of process states. If this value does not equal the expected value based on the first input parameter, the function returns the false Boolean value. It represents that the model cannot produce this part of an input history, which will stop its execution. |
| 4 | fun applyIncomeMsg(LS: LOCALSTATE, M: MESSAGE): LOCALSTATE | The function gets the current state of a process as the first input parameter and an update message from the head of the delivery queue of that process as the second parameter. The function finds the value of the variable of the message in the current status of the process and applies the update operation of the message that gets as the second parameter, which, as a result, returns the updated status of the process. |
| 5 | fun isGreatherVector(CV1: CLOCKSVECTOR, CV2: CLOCKSVECTOR, Ix: INT): BOOL | The function gets two vector clocks as the first and second input parameters and an index as the third parameter and returns a Boolean result. The first parameter represents the vector clocks of a process that delivers a message, and the second represents the vector clocks of incoming messages. The third input parameter represents the index of the process that has been sent the message. It returns a boolean value by comparing two first input vector clocks. |
| 6 | Fun isPickupEnabled (inM: INCOMEMESSAGES, P: PROCESS, CV: CLOCKSVECTOR): BOOL * INT * MESSAGE | The function gets a list of incoming messages as the first input parameter, a process number as the second input parameter, and vector clocks of that process as the third input parameter. This function returns a Boolean value, an integer value, and a message. |
| 7 | fun pickupMsg(IM: INCOMEMESSAGES, P: PROCESS, Idx: INT): INCOMEMESSAGES | The function returns the first message of the delivery queue and its index. The first input parameter of this function is the list of messages of the hold-back queue, and the second input parameter is the identity of the process that picks up a message for processing. This index is used as the third input parameter in calling function pickupMsg. Each time this function is called, it picks up a message from the head of the current process delivery queue and returns the updated list of messages after this pick-up. |
| 8 | fun updateClock(P: PROCESS, CV: CLOCKSVECTOR): CLOCKSVECTOR | Function updateClock gives a process identity and its vector clock as input parameters and returns the updated vector clock of the process as output result by the assumption of a new event in this process. This function increases the counter for the clock of the process by one. If the input vector clock is an empty list or the number of elements of this list is less than the number of the total process of DS, then the function returns an empty vector clock list. Prototype of this function is as follows: |
| 9 | fun sendMessage(MsL: MESSAGELIST, OP: OPERATION, P: PROCESS, CV: CLOCKSVECTOR): SENDMESSAGE | The function gets the list of issued messages of a process as the first input parameter, an operator as the second parameter, a process identity as the third parameter, and vector clocks of that process as the fourth parameter. This function returns the next issuing message of the process based on the operation of the second input parameter. If the operation is read from memory, no message will be issued. If the operation is a write, then the function updates the vector clocks of the process. |
| 10 | fun updateClockByMsg (MCV: CLOCKSVECTOR, RP: PROCESS, RCV: CLOCKSVECTOR): CLOCKSVECTOR | The function gives the vector clock of the delivered message as the first input parameter, the identity of the process that delivers the message, and its vector clock as the second and third input parameters. If the list of each vector clock is empty or contains fewer elements than the number of total processes of DS, then the function returns the empty list as a result. Otherwise, the function returns an updated vector clock of the message-delivering process based on the Fidge algorithm as the result. |
| 11 | fun check(OP: OPERATION, sequenceNo: INT): BOOL | The function gets an operation as the first input parameter and a global sequence number as the second parameter. If the sequence number of the operation is less than or equal to the global sequence number of the model, then the function returns a value true as a result. |
| 12 | fun initializeClocks( ): CLOCKSVECTOR | The function returns a vector of zero values with the length of the system's constituting processes. This vector is used as the initial values of the vector clocks of processes. |

| Table 3 | (continued) | |
|---|---|---|
| **Index** | **Function signature** | **Description** |
| 13 | fun propose(OP: OPERATION, LS: LOCALSTATE) | The function gets an operation of a process as the first input parameter and the current values of the process's local variables as the second parameter. If the operation is a read operation and the value is equal to the local value of that variable in the process, then the read operation is correct, and no modification is required. Otherwise, it returns the operation with the correct value of the read operation of the variable. |

The function *isPickupEnabeled* returns the first message of the delivery queue and its index. This index is used as the third input parameter in calling function *pickupMsg*. The first input parameter of this function is the list of messages of the hold-back queue, and the second input parameter is the identity of the process that picks up a message for processing. Each time this function is called, it picks up a message from the head of the current process delivery queue and returns the updated list of messages after this pick-up. Each process sends its updated vector clock and a message to the receiver. The process that delivers the message updates its vector clock by information of the vector clock of the incoming message based on the Fidge algorithm. This work is accomplished by calling the function *updateClockByMsg*.

The function *check* is used as a guard condition of transition ExecuteOperation in the low-level model of Fig. 5. This guard condition means that until any operation exists with a low sequential number, the model will never execute an operation with a greater sequential number of any other process. The function *sendMessage* is used in the inscription of the output arc of transition ExecuteOperation. If the operation is read, this transition does not add any message to the end of the list of issued messages of a process held in place OutBox and returns an unchanged list to this place. If the operation is a write operation, a new message is added to the end of the issued messages list of place OutBox.

The next section of the article discusses the analysis of the state space graph of the proposed method. The model-checking approach, which is presented in the next section, also can be considered a minor part of the materials and method. The proposed hierarchical model of the CCDS is designed to yield intended results with simple model-checking, and the heaviest part of the work has been accomplished in the modeling part.

## DISCUSSIONS AND RESULTS

This part of the article discusses the results of running the proposed model using four different examples of DH as case studies 1 to 4 for analysis. The analysis is performed by model checking of the proposed colored Petri net model *via* build-in functions of CPN tools and our customized state space analysis codes. All modelings, state space graph generation and analysis of the graph performed on an ASUS Zen Book Flip 14 laptop with Intel[(R)] Core[(TM)] i7-10510U CPU 2.30 GHz x64-based processor, 16.0 GB main memory on Windows 10 64-bit home edition operating system.

### Case studies and analysis of the state space graphs

Report 1 shows the summary report of state space generation of the proposed model of CCDS by the input DH of case study 1, which is presented in Table 1. The CPN tool

**Table 4 State space analysis functions.**

| Index | Function signature | Description |
|---|---|---|
| 1 | fun hasAnswer(): BOOL | This state-space analysis function verifies whether the input DH is the valid history of CCDS. If, during the model execution, the current operation of the running process is not valid based on the CCDS, then the proposed model stores the negative of the current step of input DH's analysis, which is available at place GlobalStep of the module CausalConsSys. A negative value at this place causes the execution of all operations of all processes to stop, and after a few steps, the model reaches a dead marking. It searches dead marking states of the stape-space graph to find nodes that the marking of the place GlobalStep is bigger than zero and returns the true as a result. If the marking of place GlobalStep is zero or negative in all dead markings of the state-space graph, it returns false. |
| 2 | fun shortestAnswer(): INT*INT | This function investigates all dead markings of the state-space graph, which marking of the place GlobalStep is positive in them. It uses the built-in function of CPN tools named NodesInPath to find the path length from the initial node of the state-space graph up to that node. The function returns the index of the first dead marking node with the shortest path length from the initial node and also returns the length of this path. |
| 3 | fun getErrorStep(): INT | When the model finds any incorrect operation, it stores the negative of the global step number of the model in the place GlobalStep. Dead markings with the lowest negative value as marking of this place show the first invalid operation in the DH. The function returns absolute values of this minimal found negative value. |
| 4 | fun proposeCorrection(): OPERATION | Like the function getErrorStep, this function finds the first dead marking node in the state-space graph, where the value of place GlobalStep has a minimum negative value. It returns the marking of place Correction of module CausalConsSys in that state-space node. |

automatically generates the model's SSG. The status of the state space graph is full, and the CPN tool has completed the generation of the state space graph in less than a second. The state space and strongly connected component (SCC) graphs have 46 nodes and 72 edges. The state space graph has eight dead markings; some presented in the report.

**Report 1: Summary report of state space of proposed model based on the input history of case study 1.**

This article presents two customized proposed functions for analyzing the state space graph. SSG's final (dead) states can be divided into two groups: (1) Dead states' markings in which the global sequence counter's value is negative. These states are not possible by the casually consistent distributed system. The absolute value of this number denotes the global step number of input DH that is not valid with CCDS. (2) Dead states that the global sequence counter place marking is bigger than zero in them. These states are reachable in causally consistent distributed systems.

Table 4 briefly describes the four developed functions for customized model checking (analysis) of the state space graph.

The first function is *hasAnswer*, and its signature is as follows: *fun hasAnswer(): BOOL*. This function does not have an input parameter and returns a Boolean result. It calls a predefined state space analysis function of CPN tools to get the list of all dead states of SSG. Then, it searches the states in the list that the global sequence number of them is non-zero. It does so by looking at the marking of the place GlobalStep if the top-level module of the proposed system model is shown in Fig. 5. If any such state exists, it returns true value. Figure 8 shows the results of running codes for analysis of the proposed model after the generation of the SSG by considering the input DH of case study 1. Calling the CPN tools's built-in function *ListDeadMarkings()* returns the list of dead states of the SSG. Calling the

**Figure 8 Result of state-space analysis of the proposed model for case study 1.**

**Table 5 History of operations of case study 2.**

| Process | Global sequence number | | | | |
|---|---|---|---|---|---|
| | 1 | 2 | 3 | 4 | 5 |
| $P_1$ | $W(x){:}4$ | $R(x){:}1$ | $W(y){:}3$ | $R(z){:}5$ | $R(x){:}6$ |
| $P_2$ | $W(x){:}6$ | | | $R(y){:}3$ | $R(x){:}1$ |
| $P_3$ | $W(x){:}1$ | $R(x){:}4$ | $W(z){:}5$ | | |

function $List.length(DeadMarkings)$ returns the number of dead states in the SSG. Our proposed function $hasAnswer()$ returns the value false as a result and shows that this input DH of case study 1, which is shown in Table 1, is not a valid image of a CCDS. The function $getErrorStep()$ returns the global step number of invalid input DH, step 5 in case study 1. Calling the function $proposeCorrection()$ shows the first required modification such that the input DH of case study 1 becomes a valid history of a CCDS. Input DH may require more corrections, and the function $proposeCorrection$ only proposes the first modification. This function proposes that the result of the read operation of process three at global step number 5 on the variable must return one instead of 0 in Table 1.

***Case study 2.*** Let us assume that the operations of processes in a DH are as shown in Table 5.

Report 2 shows the summary report of the state space graph of the case study 2. The status of the state space graph generation is full, has 27,434 nodes, and has 60,915 arcs. It was generated in 43 s. Case study 2 has a bigger state space graph than case study 1. Figure 9 shows the results of analyzing the state space graph. The call of function $hasAnswer$ returns the true value, which means that the given DH is a valid history of a CCDS.

**Figure 9 Result of state-space analysis of the proposed model for case study 2.**

Our second developed function for state space analysis is the function *shortestAnswer*. Prototype of it is: *fun shortestAnswer(): INT \* INT*. This function will be called if the result of the calling function *hasAnswer* be true. This function investigates only terminal (dead) nodes of the SSG that global sequence number of the model is not zero for them. This function returns (1) a node number from investigated nodes whose distance from the starting node of SSG (node no. 1) is less than the other nodes and (2) the length of this path. This function finds the shortest proof scenario for a given distributed snapshot by CCDS.

The result of calling function *shortestAnswer* for the distributed snapshot of case study 2 is shown in Fig. 9. The path's sequence of nodes' numbers yields by calling this function. The sequence of events can be extracted by investigating the edges of this path. Analysis shows that the shortest proof scenario for generating the DH of case study 2 comprises 31 nodes and requires 30 transition firings.

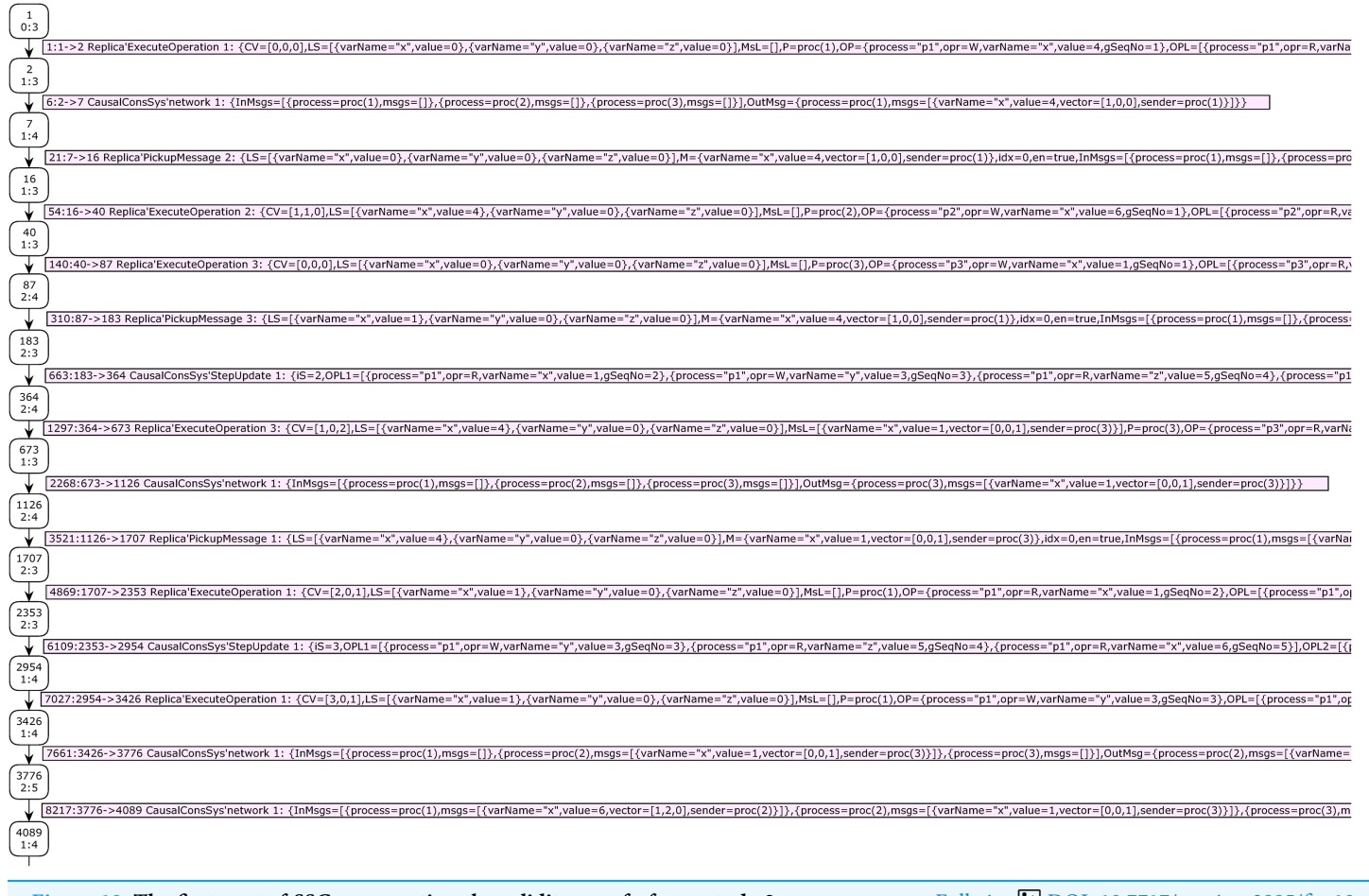

**Figure 10 The first part of SSG representing the validity proof of case study 2.**

**Report 2: Summary report of state space of proposed model based on the input history of case study 2**.

Passing through 31 nodes starting from the system's initial state makes it possible to reach a dead state that completely yields the DH of case study 2. The nodes' numbers of this path are obtained by calling the code: *if isCCDH = true then NodesInPath(1, #1 (ShortestPath)) else* []. Thirty edges of this path represent the sequence of events that produce DH of case study 2. Figures 10 and 11 show part of the SSG of case study 2 that presents the nodes of the automatically extracted shortest path. Some parts of the arcs' information, which is crucial for visualization of the proof scenario, are shown in these figures.

Figure 12 shows the manually generated proof scenario from extracted information of the state space graphs' analysis. We visualized the extracted proof scenario for the validity of case study 2 input DH *via* computation of the shortest path, as is shown in Fig. 12. Values of the Fidge's clocks of the events are shown in this figure.

***Case study 3.*** Let us assume the operations of processes in a DS is as shown in Table 6.

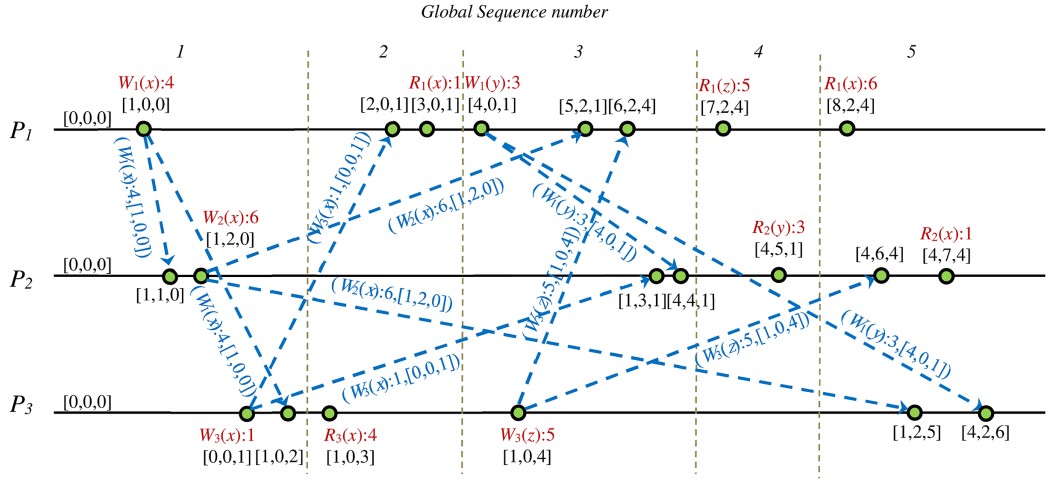

```
┌──────┐
│ 4089 │
│ 1:4  │
└──────┘
   │  9070:4089->4529 Replica'ExecuteOperation 3: {CV=[1,0,3],LS=[{varName="x",value=4},{varName="y",value=0},{varName="z",value=0}],MsL=[],P=proc(3),OP={process="p3",opr=W,varName="z",value=5,gSeqNo=3},OPL=[],S=3}
┌──────┐
│ 4529 │
│ 2:5  │
└──────┘
   │  10700:4529->5315 Replica'PickupMessage 1: {LS=[{varName="x",value=1},{varName="y",value=3},{varName="z",value=0}],M={varName="x",value=6,vector=[1,2,0],sender=proc(2)},idx=0,en=true,InMsgs=[{process=proc(1),msgs=[{varN
┌──────┐
│ 5315 │
│ 3:4  │
└──────┘
   │  13651:5315->6701 CausalConsSys'network 1: {InMsgs=[{process=proc(1),msgs=[]},{process=proc(2),msgs=[{varName="x",value=1,vector=[0,0,1],sender=proc(3)},{varName="y",value=3,vector=[4,0,1],sender=proc(1)}]},{process=proc(3)
┌──────┐
│ 6701 │
│ 2:4  │
└──────┘
   │  18524:6701->8912 Replica'PickupMessage 1: {LS=[{varName="x",value=6},{varName="y",value=3},{varName="z",value=0}],M={varName="z",value=5,vector=[1,0,4],sender=proc(3)},idx=0,en=true,InMsgs=[{process=proc(1),msgs=[{varNa
┌──────┐
│ 8912 │
│ 1:3  │
└──────┘
   │  25482:8912->11710 Replica'PickupMessage 2: {LS=[{varName="x",value=6},{varName="y",value=0},{varName="z",value=0}],M={varName="x",value=1,vector=[0,0,1],sender=proc(3)},idx=0,en=true,InMsgs=[{process=proc(1),msgs=[]},{p
┌───────┐
│ 11710 │
│ 2:3   │
└───────┘
   │  33006:11710->15222 Replica'PickupMessage 2: {LS=[{varName="x",value=1},{varName="y",value=0},{varName="z",value=0}],M={varName="y",value=3,vector=[4,0,1],sender=proc(1)},idx=0,en=true,InMsgs=[{process=proc(1),msgs=[]},
┌───────┐
│ 15222 │
│ 2:3   │
└───────┘
   │  41247:15222->18804 CausalConsSys'StepUpdate 1: {iS=4,OPL1=[{process="p1",opr=R,varName="z",value=5,gSeqNo=4},{process="p1",opr=R,varName="x",value=6,gSeqNo=5}],OPL2=[{process="p2",opr=R,varName="y",value=3,gSeqNo=4
┌───────┐
│ 18804 │
│ 3:4   │
└───────┘
   │  48321:18804->21869 Replica'ExecuteOperation 1: {CV=[6,2,4],LS=[{varName="x",value=6},{varName="y",value=3},{varName="z",value=5}],MsL=[],P=proc(1),OP={process="p1",opr=R,varName="z",value=5,gSeqNo=4},OPL=[{process="p
┌───────┐
│ 21869 │
│ 2:3   │
└───────┘
   │  53376:21869->24051 Replica'ExecuteOperation 2: {CV=[4,4,1],LS=[{varName="x",value=1},{varName="y",value=3},{varName="z",value=0}],MsL=[],P=proc(2),OP={process="p2",opr=R,varName="y",value=3,gSeqNo=4},OPL=[{process="p
┌───────┐
│ 24051 │
│ 2:3   │
└───────┘
   │  56492:24051->25369 CausalConsSys'StepUpdate 1: {iS=5,OPL1=[{process="p1",opr=R,varName="x",value=6,gSeqNo=5}],OPL2=[{process="p2",opr=R,varName="x",value=1,gSeqNo=5}],OPL3=[],S=4}
┌───────┐
│ 25369 │
│ 1:4   │
└───────┘
   │  58249:25369->26093 Replica'ExecuteOperation 1: {CV=[7,2,4],LS=[{varName="x",value=6},{varName="y",value=3},{varName="z",value=5}],MsL=[],P=proc(1),OP={process="p1",opr=R,varName="x",value=6,gSeqNo=5},OPL=[],S=5}
┌───────┐
│ 26093 │
│ 1:3   │
└───────┘
   │  59328:26093->26586 Replica'PickupMessage 2: {LS=[{varName="x",value=1},{varName="y",value=3},{varName="z",value=0}],M={varName="z",value=5,vector=[1,0,4],sender=proc(3)},idx=0,en=true,InMsgs=[{process=proc(1),msgs=[]},{
┌───────┐
│ 26586 │
│ 2:2   │
└───────┘
   │  60190:26586->27055 Replica'PickupMessage 3: {LS=[{varName="x",value=4},{varName="y",value=0},{varName="z",value=5}],M={varName="x",value=6,vector=[1,2,0],sender=proc(2)},idx=0,en=true,InMsgs=[{process=proc(1),msgs=[]},{
┌───────┐
│ 27055 │
│ 3:2   │
└───────┘
   │  60753:27055->27358 Replica'ExecuteOperation 2: {CV=[4,6,4],LS=[{varName="x",value=1},{varName="y",value=3},{varName="z",value=5}],MsL=[],P=proc(2),OP={process="p2",opr=R,varName="x",value=1,gSeqNo=5},OPL=[],S=5}
┌───────┐
│ 27358 │
│ 4:1   │
└───────┘
   │  60910:27358->27434 Replica'PickupMessage 3: {LS=[{varName="x",value=6},{varName="y",value=0},{varName="z",value=5}],M={varName="y",value=3,vector=[4,0,1],sender=proc(1)},idx=0,en=true,InMsgs=[{process=proc(1),msgs=[]},
┌───────┐
│ 27434 │
│ 5:0   │
└───────┘
```

**Figure 11  The second part of SSG representing validity proof of case study 2.**

**Figure 12  Extracted sequence diagram of events from SSG that represents validity proof of case study 2 on a CCDS.**

**Table 6 History of operations of case study 3.**

| Process | Global sequence number | | | | | | | | | |
|---|---|---|---|---|---|---|---|---|---|---|
| | 1 | 2 | 3 | 4 | 5 | 6 | 7 | 8 | 9 | 10 |
| $P_1$ | W(y):4 | | | | R(x):0 | | W(x):3 | | | |
| $P_2$ | | | R(y):4 | W(w):6 | | R(x):5 | | R(z):2 | | |
| $P_3$ | | | | R(x):5 | R(w):6 | W(z):2 | | R(x):3 | R(y):4 | |
| $P_4$ | | W(x):5 | | | | | R(y):4 | | R(z):2 | R(w):0 |

```
val DeadMarkings =
 [9999,9993,9992,998,9978,9977,9976,9966,996,9940,9939,993,9906,9905,9862,
 9861,9857,9853,9852,9851,9845,9844,9831,983,9829,9828,9827,9818,9815,9811,
 980,9794,9793,9792,9786,9785,9771,9770,977,9769,9759,9744,9743,974,9730,
 973,9729,971,9689,9688,9684,968,9679,9678,9665,9664,9630,9629,9628,962,
 9616,9615,961,9598,9554,9553,9520,9519,950,945,943,9423,9422,9418,9414,
 9413,9412,9406,9405,940,9392,9390,9389,9388,9379,9376,9372,9355,9354,9353,
 9347,9346,9332,9331,9330,9320,9305,9304,930,9291,...] : Node list
```

```
val DeadMarkings = ListDeadMarkings()            val it = 5062 : int

val IsCCDH = false : BOOL
                                                 List.length( DeadMarkings )

val IsCCDH = hasAnswer()          val ErrorStep = 10 : INT

                                  val ErrorStep = if IsCCDH = false then getErrorStep() else ~1

val Correction = [{gSeqNo=10,opr=R,process="p4",value=6,varName="w"}]
 : OPERATION ms

val Correction = if IsCCDH = false then 1` proposeCorrection() else empty
```

**Figure 13 Result of state space analysis of the proposed model for case study 3.**

Case study 3 comprises four processes compared to case studies 1 and 2, which contain only three processes. This case study comprises four variables compared to case Study 1, which contains only two variables, and case study 2, which contains three. This case study demonstrates and evaluates the proposed model's scalability. The proposed model is hierarchical, and increasing the number of processes requires increasing the instance of module "replica" in the top-level module and adding required arcs. The state space summary report of the model based on the input history of case study 3 is shown in report 3. The SSG of this case study has 23,081 nodes and 62,140 arcs. This graph is much bigger than the SSG of previous case studies. The graph was generated in 77 s, and the model has 5,062 dead markings.

**Table 7 History of operations of case study 4.**

| Proces | Global sequence number | | | | | | | | |
|---|---|---|---|---|---|---|---|---|---|
| | 1 | 2 | 3 | 4 | 5 | 6 | 7 | 8 | 9 |
| $P_1$ | W(x):3 | | | R(v):0 | | R(y):7 | | R(w):0 | |
| $P_2$ | | | | R(v):5 | W(y):7 | | R(w):9 | | R(z):0 |
| $P_3$ | | R(x):3 | W(v):5 | | | R(w):0 | R(y):7 | R(w):9 | |
| $P_4$ | | | | | R(x):0 | R(w):9 | W(z):2 | | |
| $P_5$ | | | | R(v):5 | W(w):9 | | R(y):0 | R(x):3 | R(z):2 |

**Report 3: Summary report of state space of proposed model based on the input history of case study 3.**

Figure 13 shows the result of the state space analysis. Call of function *hasAnswer* on the state space graph of case study 3 returns false, representing that the DH of case study 3 is invalid under CCDS. The result of the calling function *proposeCorrection* proposes modifying the value in DH. State space analysis shows that at step 10, the result of the read operation of process four on variable w must be 6, but in Table 6, the value zero is denoted.

***Case study 4.*** Let us assume that the operations of processes in a DH are as shown in Table 7.

DH of case study 4 is composed of five processes and five variables. Report 4 shows the summary report of the state space graph of the case study 4. The state space graph generation status is full, has been completed in 15 s, and has 17,867 nodes and 43,886 arcs. Case study 4 demonstrates the scalability of the proposed model.

Figure 14 shows the results of analyzing the state space graph. The call of function *hasAnswer* returns the true value, which means that the given DH is a valid history of a CCDS. Analysis of the state space graph by calling the function *shortestAnswer* shows that the shortest proof scenario for generating the DH of case study 4 comprises 55 nodes and requires 54 transition firings. Figure 14 displays the sequence of state space nodes that generate the proof scenario of input DH in case study 4.

**Report 4: Summary report of state space of proposed model based on the input history of case study 4.**

## Scalability of the proposed model

One important issue regarding the proposed model is its scalability. How much scalable is the proposed model? The scalability of the model can be studied from the following viewpoints: (1) scalability regarding the number of variables (data items) of the input DH, (2) scalability regarding the number of the constituting processes, and (3) scalability of model checking the state space graph of the model.

The constant model parameter *NoVars* defines the number of variables (data items) used by the input DH and, accordingly, in the model. In case studies one up to 4, the value of this parameter was 2, 3, 4 and 5, respectively. The proposed model has been investigated regarding the value of this parameter. Results showed that the proposed model is designed

```
val DeadMarkings =
  [9995,9994,9992,9990,9989,9984,9983,9976,9975,9974,9973,9971,9970,9969,9968,
   9967,9965,9962,9946,9945,9944,9943,9942,9940,9939,9938,9936,9935,9933,9931,
   9929,9928,9927,9926,9925,9924,9923,9922,9920,9919,9918,9917,9916,9914,9913,
   9912,9910,9909,9907,9905,9903,9886,9885,9884,9883,9882,9881,9880,9879,9877,
   9876,9875,9874,9872,9871,9869,9867,9865,9863,9861,9858,9855,9854,9853,9851,
   9850,9849,9845,9841,9840,9839,9837,9836,9835,9834,9830,9828,9827,9825,9820,
   9819,9817,9815,9813,9808,9807,9805,9802,9801,9796,...] : Node list
```

```
val DeadMarkings = ListDeadMarkings()        val it = 7223 : int

val IsCCDH = true : BOOL
                                             List.length( DeadMarkings )
```

```
val IsCCDH = hasAnswer()
val ShortestPath = (17867,55) : INT * INT
```

```
val ShortestPath = if IsCCDH = true then  shortestAnswer() else (~1,~1)

val it = [9] : INT ms
```

```
if IsCCDH = true then
Mark.CausalConsSys'GlobalStep 1( #1 (ShortestPath) )
else []
```

```
val it =
  [1,2,3,6,17,32,46,56,63,74,101,153,232,281,380,480,551,590,605,621,642,693,
   799,959,1258,1557,2137,2947,3968,5150,6382,7566,8583,9324,9783,10055,10255,
   10561,11019,11635,12410,13302,14202,15042,15734,16285,16740,17132,17452,
   17659,17755,17817,17847,17862,17867] : Node list
```

```
if IsCCDH = true then
NodesInPath(1, #1 (ShortestPath) )
else []
```

**Figure 14 Result of state space analysis of the proposed model for case study 4.**

such that the value of this parameter does not affect the size of the model's state space graph. This means that the proposed model is highly scalable in terms of the number of variables in input DH. Increasing the number of variables increased the state space generation time a little. It increases the generation of the state space graph because of the increase in the size of the color sets, such as the list that we used to store values of local copies of the variables in each process and linear searches in these lists. The time of generation of the state space graph has *an O(1) relationship regarding the number of variables in the input DH and* the model. The model parameter *NoVars* is used in the functions findVariableIndex, checkRead, updateLocalState, and applyIncomeMsg and initializeClocks used in the Replica module, as shown in Figs. 6 and 7.

The constant model parameter *NoProcs* defines the number of constituting processes used by the input DH and, accordingly, in the model. The proposed model is designed to be scalable in terms of the number of constituting processes. Although the scalability regarding the number of processes is not only *via* changing the initial marking of the

proposed model, a minor modification in the top-level module is also required. The proposed model has a hierarchical design, which helps to increase the number of constituting processes by increasing the substitution transition of the Replica module. Case studies 2 and 3 are designed with three processes, case study 3 with four, and case study 4 with five processes. Figure 5 shows the top-level module of case study 1, which contains three processes. The guard condition of the transition *StepUpdate* in Fig. 5 requires a variable number of arguments regarding the scalability issue of the proposed model that is not supported in standard ML and CPN ML. Fold structure constituting two functions *fold* and *step0*, along with two functions *$* and *a* have been defined to support the variable number of parameters for calling the function *minHdStep*. All of these functions are visible in the structure chart of Fig. 7. This technique increases the proposed model's scalability regarding the number of constituting processes. Increasing the number of processes increases the size of the model's state space graph and its generation time.

Write operations will be groupcasted to other model processes, and considering the variable delay of message transmissions increases the size of the model's state space graph. Read operations only increase the size of the state space graph by a small amount. By increasing the size of the state space graph, it is evident that the processing time of some of the model checkings (analysis) of this graph will be increased. The processing order of some model checkings is in quadratic order, based on the number of nodes and/or arcs of the state space graph. Therefore, increasing the state space graph's size will increase the time of model-checking.

Different approaches can be used to alleviate this limitation. The first approach is considering reasonable assumptions in the model. We can consider a limit to the delay in message transmission. This limiting assumption decreases the size of the state space graph of the model. The second approach is changing the CPN tools software so that we can combine the generation of the state space graph and do model chacking concurrently. Model checking is usually done after the generation of the state space graph of the proposed model. Doing model checking simultaneously with generating the state space graph significantly decreases the analysis time. When we find that this input DH is not valid during the generation of the state space graph, we can stop the generation of the remaining part and report the evidence.

Some analyses do not require that we backtrack the traversed path of the state space graph. As the third approach, if the current node of the state space graph is valid, we are not required to keep up with the previous nodes. This approach decreases the processing time and memory usage. However, this approach is not applicable in all types of analysis but in some applications considering the appropriate revision of the proposed model to support it. Applying the second and third approaches requires access to the CPN tools' source code and changing it to support them.

## Limitations of the proposed model

The proposed model is designed to be as simple as possible. The main reason is avoidance of the state space explosion problem. The most important limitations of the proposed model are as follows: (1) the read/write operations of the input DH have a simple style and

must be divided into separate steps to make its process easier. (2) The user needs to initialize the model's parameters NoVars and NoProcs, according to the input DH. (3) Increasing the number of processes in the model requires adding substitution transition of module "Replica," adding the required arcs, and revising the guard condition of the transition StepUpdate in the top-level module shown in Fig. 5. The authors used the fold techniques to support variable input parameters to function minHdStep in the guard condition of this transition. (4) By increasing the number of processes and the write operations in the input DH, the size of the state space graph will increase extensively. This phenomenon is expected and is the inherent behavior of the formal methods based on generating the state space graph. One of the best approaches to deal with this problem is combining the process of generating the state space graph and concurrently doing our customized model checking. (5) The analysis of the state space graph generates the proof scenario if the input DH is valid under CCDS. The developed function extracts the proof scenario automatically *via* customized model checking, but it has a lot of detailed information that requires deep familiarity with color sets and marking of the state space nodes. Automatic generation of a message sequence chart helps immensely in understanding the proof scenario in a shorter time without the requirement of understanding the details of the model design. (6) It is possible to design the model such that limitation three, mentioned before, has been eliminated. It causes more scalability of the model, such that to increase the number of processes, the model's top-level module will not be required to be modified. Increasing the number of processes can be accomplished only by modification of the input markings related to the input DH. However, developing such a model has some issues: (1) the complexity of the model's color sets increases extensively and understanding the state space nodes' marking becomes more complicated. (2) All model functions and state space analysis must be revised accordingly. (3) The processing time of the state space generation increases. Nevertheless, it causes a decrease in the number of palaces in the model and, therefore, decreases the size of the state space graph.

## CONCLUSIONS

Formal modeling and verification are important because they can prove complex systems' properties. Analyzing systems with large SSG is hard and requires appropriate tools for automatic analysis. Colored Petri net with high modeling capabilities can be used in various application areas. One of the most used algorithms for implementing the causal consistency in DS, which is based on the Fidge's clocks, is studied in this article. The hierarchical model of a CCDS with three constituting machines (processes) based on the mentioned algorithm is introduced in this article. Input DH is fed to the model as the initial marking.

This article presents and uses three effective modeling mechanisms against the state space explosion problem. The first mechanism was using ML codes in the form of functions. It decreases the number of transitions and places in the model, and their number directly relates to the size of the SSG of the model. CPN tool is sensitive to the sequence of list elements, and different substitutions of list elements are considered as

different system states. In this article, lists are mostly used to represent sets. We used the list structure instead of the set, and elements in the list were also sorted to decrease the size of the state space graph. Therefore, sorting lists' items was the second mechanism against the state space explosion problem. The third mechanism was stopping the execution of the model as soon as it visited a state that CCDS could not produce.

Four new functions for analyzing the SSG were presented. The first one searches the dead markings of the SSG to find whether the input DH is valid. The second function searches the SSG to extract the shortest path for extracting the simplest proof scenario. The third function extracts the global sequence number of the step that contains the first incorrect read operation value in the input DH. The last function proposes the correct value of incorrect read operation in the DH.

Four case study DHs are presented in the article, and functions for analyzing the SSG of the model and extracting correctness proof of valid DHs are presented. The proposed model permits the user to give a DH of a DS to the model as input and, by automatic state space analysis of the CCDS model, finds whether the given DH is valid by CCDS or not. If valid, the user can automatically extract the shortest proof scenario that CCDS can produce.

In case studies 2 and 4, analyzing the big SSG for computing the shortest path is time-consuming compared to the time of SSG generation, which took a few minutes. This delay is expected because of the large size of the SSG and the extensive search required to find the shortest path. Based on the success of the proposed model, colored Petri net models of DSs that guarantee other consistency models of DS will be studied.

## FUTURE WORKS

In the current work, model-checking of the proposed work verifies whether the current input DH can be a valid history of a CCDS or not. If valid, it extracts a scenario of events from the state space graph of the system's model such that a CCDS can generate the input DH. If it is valid, the output of model checking is accurate and contains detailed information on events in the DS. It contains a path of the state space graph from the system's initial state ending in one of its final states that represents how the input DH can be generated in CCDS. Such validation proof has detailed information and is time-consuming and complicated for users to analyze. The authors manually generated a message sequence chart of this validation proof to show a scenario of CCDS that can generate input DH, as shown in Fig. 12. CPN tools have simple basic facilities for generating message-sequence charts *via* adding ML codes to the model. As the first future work, we would like to revise the proposed model to automatically generate the message sequence chart of the extracted proof scenario of model-checking that can be easily understandable by users of this article's proposed model.

Many consistency models have been developed for distributed systems. Researchers develop customized versions of the classical consistency models for new applications on extensions of distributed systems like wireless sensor actuator networks or the Internet of Things. Newly developed customized consistency models require verification of their key characteristics. Developing colored Petri nets of classical and new consistency models like

FIFO and sequential consistency models and model checking them is our second work in the future.

If we consider a long DH or DS consists of many processes, or both of these cases happen, then the state space graph of the proposed model will be huge. It causes the state space explosion problem. By increasing the length of DH and/or the number of processes, the size of the state space graph grows exponentially. Model-checking can not be accomplished because generating state space graphs and running model-checking algorithms takes a long time. Based on the memory, processing power of the computer, and time limitation, most of the time, the state space graph will be partial.

One approach to overcome this problem is model-checking during the generation of the state space graph, eliminating the parts of the state space graph that have been analyzed by model-checking and keeping it, which is no longer required. This approach only applies in a few studies and can be used in current research. In most cases, when a message is delivered to the receiver, we can study by model-checking whether the input DH is consistent with the definition of CCDS or not up to this point. If it is not, then continuing the study of the current path (scenario) of the state space graph is not required, and we can stop the generation of part of the state space graph rooted from the current node. Beyond that, when we reach a node of the state space graph that proves the current input DH is a valid output of CCDS, we can stop the generation of the remaining parts of the state space graph. Applying this approach requires developing the revised version of CPN tools, state space generation, and model-checking methods that support this feature. Our third proposed future work is applying customized state space generation and model-checking to alleviate the state space explosion problem.

As the fourth future work, to increase the model's scalability, it is possible to revise the model to automatically determine the number of processes in the input DH and support a variable number of processes without modifying the proposed model's top-level module.

## ACKNOWLEDGEMENTS

The authors would like to thank respected reviewers for their directions in improving the quality of the article.

### Funding

The authors received no funding for this work.

### Competing Interests

The authors declare that they have no competing interests.

### Author Contributions

- Khalid Amjed Mohammed Alsaegg conceived and designed the experiments, performed the experiments, analyzed the data, performed the computation work, prepared figures and/or tables, and approved the final draft.

- Saeid Pashazadeh conceived and designed the experiments, performed the experiments, analyzed the data, performed the computation work, prepared figures and/or tables, authored or reviewed drafts of the article, and approved the final draft.
- Mina Zolfy Lighvan conceived and designed the experiments, analyzed the data, authored or reviewed drafts of the article, and approved the final draft.

## Data Availability

The code is available in the Supplemental File.

## Supplemental Information

Supplemental information for this article can be found online at http://dx.doi.org/10.7717/peerj-cs.2995#supplemental-information.

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
