# Peer review of "Formal modeling of a causal consistent distributed system and verification of its history via model checking using colored Petri net"

_PeerJ Computer Science, doi:10.7717/peerj-cs.2995_

## Round 0.1 · original submission · Major Revisions

The reviewers found the paper interesting and well-written, they also highlight the importance of the topic. Still, the paper needs improvement before publication.

Please address the following points when revising the paper, in addition to all comments from the reviewers.

[1] Extend the related work discussion and consider more recent literature (after 2019).

[2] Improve the presentation by numbering sections and moving figures/tables in the text (now in the appendix).

[3] Add a discussion on the limitations of the approach (in particular wrt scalability) and on possible future research directions.

Reviewer 1 ·

Basic reporting

The submitted manuscript is interesting and the topic is up to date. Model checking is a promising formal verification technology that is applied in many domains.

I had no problems with understanding the paper, English language quality is for me acceptable.

All figures are clearly presented.

The structure of the paper is appropriate. I strongly suggest to add numbering to the sections.

Regarding the literature references, it is quite outdated. The newest references is from 2019, making the presented state of the art not representative. Moreover, as the authors deal with model checking of coulored Petri nets, I suggest to briefly present the state of art of model checking of different types of Petri nets, there are many recent publications on that topic, to name just some of them:
1. Wang, S., Yang, R., Yu, W., Ding, Z., & Jiang, C. (2024). Model Checking of ω-Independent Unbounded Petri Nets for an Unbounded System. IEEE Transactions on Computational Social Systems.
2. Grobelna, I., & Szcześniak, P. (2022). Model Checking Autonomous Components within Electric Power Systems Specified by Interpreted Petri Nets. Sensors, 22(18), 6936.
3. Yang, R., Ding, Z., Guo, T., Pan, M., & Jiang, C. (2022). Model checking of variable Petri nets by using the Kripke structure. IEEE Transactions on Systems, Man, and Cybernetics: Systems, 52(12), 7774-7786.

Experimental design

The motivation of the research is clear. So it is the research methodology.

Validity of the findings

Conclusions are well stated. Plans for the future should be added to the last section of the manuscript.

Additional comments

The introduction should be more supported by existing literature.

As the sections are not numbered, it is harder to follow the paper. Usually, it is expected to find a summary of paper’s structure at the end of the introduction.

I suggest to move the section with Related work just after the introduction.

Algorithm 1 Pseudo Code of Causal ordering using vector timestamps is missing. Appendix is not placed at the end of the paper. In general, I am not sure whether it is required by the template, but I found it difficult to follow the paper when the graphics are placed at the end of it. When providing a revised version, please put them in the correct place.

I also missed the limitations of the proposed approach.

Reviewer 2 ·

Basic reporting

1. BASIC REPORTING
Clear, unambiguous, professional English language used, but some terminology did not define, for example: 112 the correctness of a distributed algorithm; 35 verification.

Introduction suggests that this article is intended for 74 graduate students, background shows well known context 174 Concept of causal consistency; 291 Formal definition of colored Petri net.
Literature well referenced and relevant, but there are no references about future research direction.

Structure conforms to PeerJ standards, discipline norm, or improved for clarity.
Figures are relevant, good quality, well labeled and described.
Raw data supplied

Experimental design

2. EXPERIMENTAL DESIGN
The research provides educational material for laboratory work and for studying distributed systems.

Research question well defined. It is stated how the research fills an identified knowledge gap.

Investigation performed to an education standard.
Methods described with sufficient detail and information to replicate.

Validity of the findings

3. VALIDITY OF THE FINDINGS

All underlying data have been provided; but there is no information about system behavior for 4 and more devices, no suggestions for model scalability and future work. I recommend to study reenterable models.

The results only show how the model has been improved against the state-space
604 explosion problem. Hierarchical model, ML functions are presented like effective modeling mechanisms, but it is a usual part of complex system model.

Additional comments

This paper provides a very good educational material for laboratory work and for studying distributed systems. But without a model scalability, definition and proof of correctness it is not a research paper.

---

## Round 0.2 · accepted · Accept

The reviewers are satisfied with the revised version of the work. The authors addressed all the concerns highlighted in the previous round of review, and the paper is now ready for publication.

Reviewer 1 ·

Basic reporting

I appreciate the efforts of the authors to improve paper’s quality. I have no other comments.

Experimental design

acceptable

Validity of the findings

acceptable